# Dual-targeting CRISPR-CasRx reduces *C9orf72* ALS/FTD sense and antisense repeat RNAs in vitro and in vivo

Liam Kempthorne[1,2,12], Deniz Vaizoglu[1,2,12], Alexander J. Cammack[1,2,12], Mireia Carcolé [1,2], Martha J. Roberts [1,2], Alla Mikheenko[1,2], Alessia Fisher[1,2], Pacharaporn Suklai[1,2], Bhavana Muralidharan[1,2,3], François Kroll [4], Thomas G. Moens[5], Lidia Yshii[5], Stijn Verschoren[5], Benedikt V. Hölbling[1,2], Francisco C. Moreira [6], Eszter Katona[1,2], Rachel Coneys[1,2], Paula de Oliveira[1,2], Yong-Jie Zhang[7], Karen Jansen[7], Lillian M. Daughrity [7], Alexander McGown[8], Tennore M. Ramesh[8], Ludo Van Den Bosch[5], Gabriele Lignani [6], Ahad A. Rahim[9], Alyssa N. Coyne [10,11], Leonard Petrucelli[7], Jason Rihel [4] & Adrian M. Isaacs [1,2] ✉

The most common genetic cause of frontotemporal dementia (FTD) and amyotrophic lateral sclerosis (ALS) is an intronic $G_4C_2$ repeat expansion in *C9orf72*. The repeats undergo bidirectional transcription to produce sense and antisense repeat RNA species, which are translated into dipeptide repeat proteins (DPRs). As toxicity has been associated with both sense and antisense repeat-derived RNA and DPRs, targeting both strands may provide the most effective therapeutic strategy. CRISPR-Cas13 systems mature their own guide arrays, allowing targeting of multiple RNA species from a single construct. We show CRISPR-Cas13d variant CasRx effectively reduces overexpressed *C9orf72* sense and antisense repeat transcripts and DPRs in HEK cells. In *C9orf72* patient-derived iPSC-neuron lines, CRISPR-CasRx reduces endogenous sense and antisense repeat RNAs and DPRs and protects against glutamate-induced excitotoxicity. AAV delivery of CRISPR-CasRx to two distinct *C9orf72* repeat mouse models significantly reduced both sense and antisense repeat-containing transcripts. This highlights the potential of RNA-targeting CRISPR systems as therapeutics for *C9orf72* ALS/FTD.

Amyotrophic lateral sclerosis (ALS) and frontotemporal dementia (FTD) are two inexorable neurodegenerative disorders that exist on a common disease spectrum, characterised by motor impairment and behavioural and language deficits, respectively[1]. The most common

genetic cause of both ALS and FTD is a $G_4C_2$ hexanucleotide repeat expansion in intron 1 of chromosome 9 open reading frame 72 (*C9orf72*)[2,3]. Healthy individuals typically harbour up to 30 $G_4C_2$ repeats, whereas those with *C9orf72*-related ALS and FTD (C9 ALS/

[1]UK Dementia Research Institute at UCL, London WC1E 6BT, UK. [2]Department of Neurodegenerative Disease, UCL Queen Square Institute of Neurology, London WC1N 3BG, UK. [3]Institute for Stem Cell Science and Regenerative Medicine, Bangalore 560065, India. [4]Department of Cell and Developmental Biology, University College London, London WC1E 6BT, UK. [5]VIB-KU Center for Brain and Disease Research, Leuven 3001, Belgium. [6]Department of Clinical & Experimental Epilepsy, UCL Queen Square Institute of Neurology, London WC1N 3BG, UK. [7]Department of Neuroscience, Mayo Clinic, Jacksonville, FL 32224, USA. [8]Sheffield Institute for Translational Neuroscience, University of Sheffield, Sheffield S10 2HQ, UK. [9]UCL School of Pharmacy, University College London, London WC1N 1AX, UK. [10]Department of Neurology, Johns Hopkins University, Baltimore, USA. [11]Brain Science Institute, Johns Hopkins University, Baltimore, USA. [12]These authors contributed equally: Liam Kempthorne, Deniz Vaizoglu, Alexander J. Cammack. ✉e-mail: a.isaacs@ucl.ac.uk

FTD) may have hundreds to thousands of $G_4C_2$ repeats[4]. These repeat-containing regions are bidirectionally transcribed into both sense ($G_4C_2$) and antisense ($C_4G_2$) repeat-containing transcripts[5–8]. Both transcripts undergo repeat-associated non-AUG (RAN) translation, leading to the production of five distinct species of dipeptide repeat proteins (DPRs): polyGA, polyGP and polyGR from the sense strand and polyGP, polyPA and polyPR from the antisense strand[6,7,9–11].

To date, therapeutic strategies to target C9 ALS/FTD repeat RNAs have focused on targeting the sense $G_4C_2$ repeat transcripts[12–16], however, there is a considerable and growing body of evidence that supports the contribution of the antisense strand to C9 ALS/FTD disease pathogenesis[17–21]. Additionally, recent failures in clinical trial of two antisense oligonucleotides (ASOs) targeting *C9orf72* sense repeat-containing transcripts (BIIB078; clinicaltrials.gov: NCT03626012 and WVE-004; clinicaltrials.gov: NCT04931862), despite showing target engagement and lowering of the sense transcript, highlight the importance of developing therapeutic approaches that can also target *C9orf72* antisense repeat RNA transcripts[22,23].

Clustered regularly interspaced short palindromic repeat (CRISPR) RNAs and CRISPR-associated (Cas) proteins are part of the adaptive immunity of bacteria[24]. The recent discovery of Type VI Cas13 effectors with ribonuclease activity represent a breakthrough in RNA interference systems[25–28]. Cas13 effectors have been shown to exhibit dual ribonuclease activity in mammalian systems, with the ability to mature their own guide RNA (gRNA) arrays from a CRISPR locus as well as degrade targeted RNA strands via higher eukaryotes and prokaryotes nucleotide-binding (HEPN) domains[29]. This dual ribonuclease activity, combined with the smaller size of some Cas13 effectors (such as *Rfx*Cas13d at 967 amino acids)[28], suggest CRISPR-Cas13 systems could be utilised as therapeutics that can be packaged into an adeno-associated virus (AAV) vector and targeted to multiple human RNA sequences[30–32]. We therefore aimed to exploit CRISPR-Cas13 systems to target both the sense and antisense repeat RNA species in *C9orf72* ALS/FTD in a single therapeutic design.

## Results

### CRISPR-Cas13d(Rx) reduces *C9orf72* sense and antisense repeat RNA and DPRs in HEK cells

To determine whether CRISPR-Cas13 systems could efficiently degrade *C9orf72* repeat RNAs, we designed 30 nucleotide (nt) gRNAs to target sense $G_4C_2$ repeat-containing transcripts. We tiled the guides every 10 nucleotides directly upstream of the $G_4C_2$ repeat (Fig. 1a). Of these guides we selected those with the lowest homology to other sequences in the human transcriptome and lowest RNA secondary structure

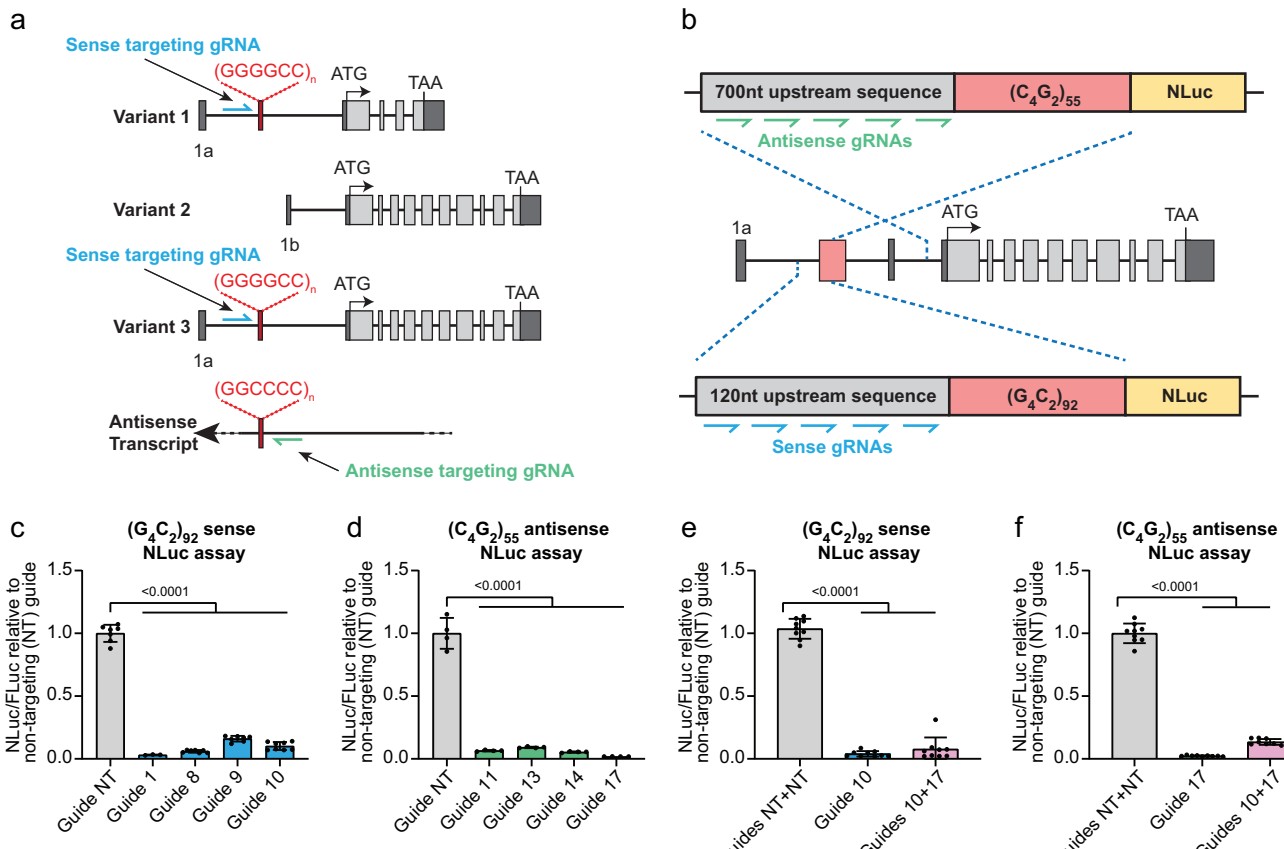

**Fig. 1 | CRISPR-CasRx can effectively lower sense and antisense repeat-containing transcripts and DPRs and prevent polyGR and polyPR formation in HEK293T cells. a** Strategy for targeting sense and antisense *C9orf72* transcripts with gRNAs. Sense gRNAs target repeat-containing variants 1 and 3 but not variant 2. **b** Sense and antisense NLuc reporter assay designs. **c** Sense NLuc assays testing single plasmids expressing both CasRx and sense guides. **d** Antisense NLuc assays testing single plasmids expressing both CasRx and antisense guides. Data in (**c, d**) are shown as mean ± S.D. n = 2 biological repeats (all technical repeats shown on graph), one-way ANOVA with Dunnett's test for post-hoc analysis. **e, f** Single vectors expressing a single guide (guide 10 for sense targeting or guide 17 for antisense targeting) or both guides 10 and 17 were used in our sense and antisense NLuc reporter assays. CasRx with sense and antisense targeting array can effectively reduce both **e** sense and **f** antisense *C9orf72* DPR levels to a similar degree to single guide expressing plasmids indicating effective guide array maturation and multi-target engagement. All NLuc data normalised to FLuc and non-targeting (NT+NT) guide. Data in (**e, f**) shown as mean ± S.D. n = 3 biological repeats (all technical repeats shown on graph), one-way ANOVA with Dunnett's test for post-hoc analysis. Source data are provided as a Source Data file.

scores, to provide 6 gRNAs to take forward for experimental analysis. We focused on the region upstream of the repeat, rather than the $G_4C_2$ sequence itself due to the very high GC content of the repeat sequence and the presence of $G_4C_2$ repeats elsewhere in the genome. In addition, *C9orf72* has three transcript variants, with the repeat expansion being present in the pre-mRNA of variants 1 and 3, but within the promoter of variant 2, thereby excluding the repeat from variant 2 pre-mRNA (Fig. 1a). Targeting gRNAs upstream of the $G_4C_2$ repeat in intron 1 allows us to target only the repeat-containing transcripts (variants 1 and 3) whilst sparing variant 2 pre-mRNA (Fig. 1a).

To test these gRNAs we utilised a NanoLuciferase (NLuc) reporter assay that we previously developed to measure sense *C9orf7*2 repeat levels (Fig. 1b)[33]. Our sense NLuc reporter consists of 92 pure $G_4C_2$ repeats with 120 nucleotides of the endogenous *C9orf72* upstream sequence followed by NLuc in frame with polyGR. As the NLuc reporter has no ATG start codon, the NLuc signal is a readout of RAN-translated polyGR. We first tested two Cas13 subtypes previously shown to efficiently target mammalian transcripts, Cas13b from *Prevotella sp. P5-125*[26] and Cas13d from *Ruminococcus flavefaciens* strain XPD3002 (CasRx)[27]. While Cas13b could effectively target the sense repeat-containing transcripts, with up to 75% reduction in NLuc signal (Supplementary Fig. S1a), CasRx achieved a 99% reduction in NLuc signal with four of the six guides tested (guides 1, 8, 9, and 10) (Supplementary Fig. S1b). These data show that CasRx is more efficient than Cas13b at reducing *C9orf72* sense repeat transcripts. Importantly, in addition to the superior targeting efficiency, CasRx is 160 amino acids smaller than Cas13b, and thus is small enough to facilitate packaging, along with a gRNA array, into adeno-associated viruses (AAVs). This striking effectiveness of CasRx to target the sense repeat-containing transcripts, combined with its smaller size, led us to move forward with CasRx. We additionally tested an enzymatically dead CasRx (dCasRx) that binds to the target transcript but lacks ribonuclease activity to degrade it[27], as we hypothesised that binding of CasRx to repeat-containing transcripts may be sufficient to prevent translation of the repeat sequence. However, dCasRx did not reduce NLuc levels, suggesting the binding of dCasRx to the transcript is not sufficient to impair RAN translation (Supplementary Fig. S1c). These data show that CasRx is very effective at reducing overexpressed *C9orf72* sense repeat RNAs, and that its ribonuclease activity is required to prevent DPR production.

As CRISPR-CasRx was very effective at reducing *C9orf72* sense repeat RNAs, we next tested whether it could also target antisense repeat-containing transcripts. We first developed an antisense NLuc reporter consisting of 55 pure $C_4G_2$ repeats and 700 nucleotides of the endogenous *C9orf72* sequence upstream of the repeat expansion in the antisense strand with NLuc in frame with polyPR (Fig. 1b). We tested 4 gRNAs targeting upstream of the antisense $C_4G_2$ in our antisense NLuc assay. These initial gRNAs were targeted as close to the $C_4G_2$ repeats as possible due to the unknown antisense transcriptional start site. All tested guides reduced the NLuc signal >70% with guide 11 achieving the greatest reduction of 89% (Supplementary Fig. S1d) despite the high GC-content (~80%) of the targeted region adjacent to the antisense repeats. Finally, in a parallel approach, we used repeat-targeted RNA fluorescent in situ hybridisation (FISH) to directly confirm *C9orf72* transcript lowering. Indeed, both sense and antisense-targeted gRNAs were able to reduce *C9orf72* repeat RNA levels to near-background levels (Supplementary Fig. S1e–h). Thus, CasRx is able to robustly degrade sense and antisense *C9orf72* repeat RNAs and prevent DPR accumulation in vitro.

## Dual targeting CRISPR-CasRx effectively reduces *C9orf72* sense and antisense repeat RNAs simultaneously in HEK cells

In these preceding experiments our gRNAs were expressed without the need for CasRx to mature a pre-gRNA prior to target recognition and binding. As the gRNA in a therapeutic vector would be in a pre-gRNA configuration, we next investigated whether CasRx could mature a pre-gRNA and maintain targeting efficiency. We cloned our gRNAs into a pre-gRNA expression vector where the gRNA spacer sequence is flanked by two direct repeats (DRs) and therefore must be matured prior to target engagement. Testing these pre-gRNAs in our sense NLuc assay confirmed that CasRx can mature a pre-gRNA and still very effectively target the sense repeat-containing transcripts (Supplementary Fig. S1i). In addition, when CasRx matures a pre-gRNA array, ~8 nt of the guide sequence can be removed along with 6 nt of the DR to form a mature gRNA. Therefore, we also tested 22 nt gRNA versions of our most efficacious 30 nt gRNAs, with 8 nt removed from the 3′ end of the guide sequence. In our NLuc reporter assays these 22 nt gRNAs successfully targeted the sense and antisense repeat-containing transcripts (Supplementary Fig. S1j, k).

We then cloned our most efficacious sense and antisense-targeting gRNAs into a single lentiviral vector also expressing CasRx. In our initial NLuc reporter assays (Supplementary Fig. S1a–d) the gRNAs were supplied as a separate plasmid, in a 5:1 molar ratio to CasRx. These single gRNA-CasRx expressing plasmids show that a 1:1 ratio of gRNA to CasRx, expressed from the same vector, is still sufficient to effectively target the *C9orf72* sense and antisense repeat-containing transcripts (Fig. 1c, d). Here we also tested a new antisense-targeting gRNA (guide 17) that targets further from the repeat sequence (<200 bp upstream of the antisense repeats, where the sequence is less GC-rich) than the other antisense guides previously tested. We found guide 17 to be the most efficacious gRNA in our antisense NLuc assay, reducing polyPR-NLuc to background levels (Fig. 1d). We therefore used this antisense-targeting gRNA moving forward.

Having established our best performing sense and antisense targeting guides, we next produced a dual targeting lentiviral construct containing CasRx and sense targeting gRNA 10 and antisense targeting gRNA 17 in a pre-gRNA array. We then compared this dual-targeting construct to our single guide constructs in our sense and antisense NLuc reporter assays. The dual gRNA expressing plasmid effectively reduced polyGR-NLuc and polyPR-NLuc levels to a comparable degree to plasmids expressing individual mature gRNAs (Fig. 1e, f). These data show effective gRNA array maturation and multi-target engagement of CasRx with sense and antisense *C9orf72* repeat RNAs in HEK cells.

## CRISPR-CasRx effectively targets sense and antisense *C9orf72* repeat RNAs in patient-derived iPSC-neurons

Following the robust targeting efficiency of CasRx in our HEK293T transient overexpression model, we wanted to determine whether these gRNAs and CasRx can successfully target endogenous *C9orf72* sense and antisense repeat transcripts in *C9orf72* ALS/FTD patient-derived iPSC-neurons. We used piggyBac-mediated integration to insert a doxycycline (dox)-inducible NGN2 cassette into iPSCs for three independent *C9orf72* patient lines to allow rapid differentiation into cortical-like neurons[34,35]. We confirmed that after 5 days of dox treatment these cells expressed neuronal markers using immunocytochemistry and RT-qPCR (Supplementary Fig. S2). We transduced these dox-inducible *C9orf72* patient NGN2 iPSC lines with our CRISPR-CasRx and gRNA expressing lentiviruses. Transduced neuronal cells were isolated with FACS after 5 days of differentiation for target engagement assessment (Fig. 2a) using custom-designed RT-qPCR assays for measuring sense and antisense repeat-containing transcripts (Fig. 2b). Our sense targeting gRNAs 8 and 10 both significantly reduced sense repeat transcript levels by ~40%, dependent on the cell line used, relative to the non-targeting gRNA control (Fig. 2c). Similarly, our antisense targeting gRNAs 11 and 17 successfully reduced the antisense repeat-containing transcripts in all three *C9orf72* ALS/FTD iPSC-neuron lines by ~32% and ~73% respectively (Fig. 2d). As predicted, there was no reduction in *C9orf72* variant 2, which does not contain the repeats within its pre-mRNA (Fig. 2e). We also performed RT-qPCR analysis for CasRx expression in these samples (Supplementary

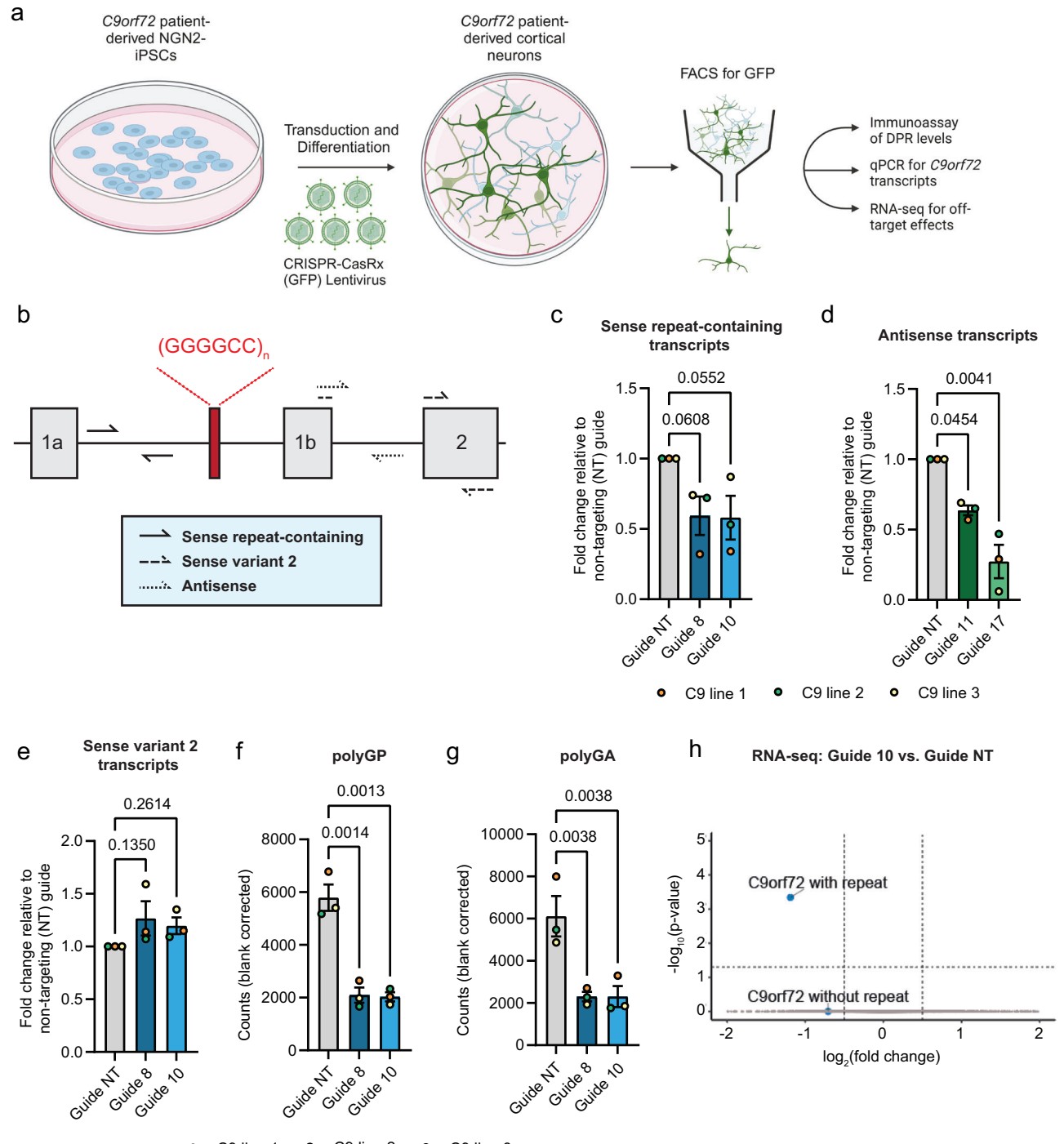

**Fig. 2 | CRISPR-CasRx reduces endogenous sense and antisense repeat-containing *C9orf72* RNA and DPRs in patient-derived iPSC-neurons without lowering variant 2 or causing detectable off-target effects. a** Experimental setup to determine target engagement in patient-derived iPSC-neurons. Created in BioRender. Cammack, A. (2024) https://BioRender.com/z76w550. **b** Schematic of RT-qPCR primers used to quantify *C9orf72* transcripts. **c**–**e** RT-qPCRs for *C9orf72* transcripts in neurons treated with CRISPR-CasRx + gRNA lentiviruses. Data presented as fold change compared to non-targeting (NT) gRNA. **f, g** Levels of polyGP and polyGA in iPSC-neurons treated with sense targeting gRNAs 8 and 10, shown as fold-change compared to average of non-targeting (NT) gRNA treatments across C9 lines. Data in (**b**–**g**) shown as mean ± S.D, n = 3 biological replicates (i.e.

individual C9 lines), each shown as different colour data points (orange, green, and yellow). Experiments were performed on three separate inductions of C9 line 1 and in one induction each of C9 lines 2 and 3. p-values calculated with two-way ANOVA and Dunnett's test for post-hoc analysis. **h** Volcano plot of DESeq2 analysis showing DEGs between neurons treated with sense-targeting guide 10 compared to non-targeting (NT) gRNA with *C9orf72* transcripts grouped by those that contain intron1 and the repeats, and those that do not. Dotted lines indicate thresholds for fold change on x-axis (|log₂FoldChange|>0.5) and *p* value on y-axis (adjusted p < 0.05, two-sided Wald test, p-values are adjusted for multiple testing using Benjamini-Hochberg method). n = 3 independent inductions of C9 line 1. Source data are provided as a Source Data file.

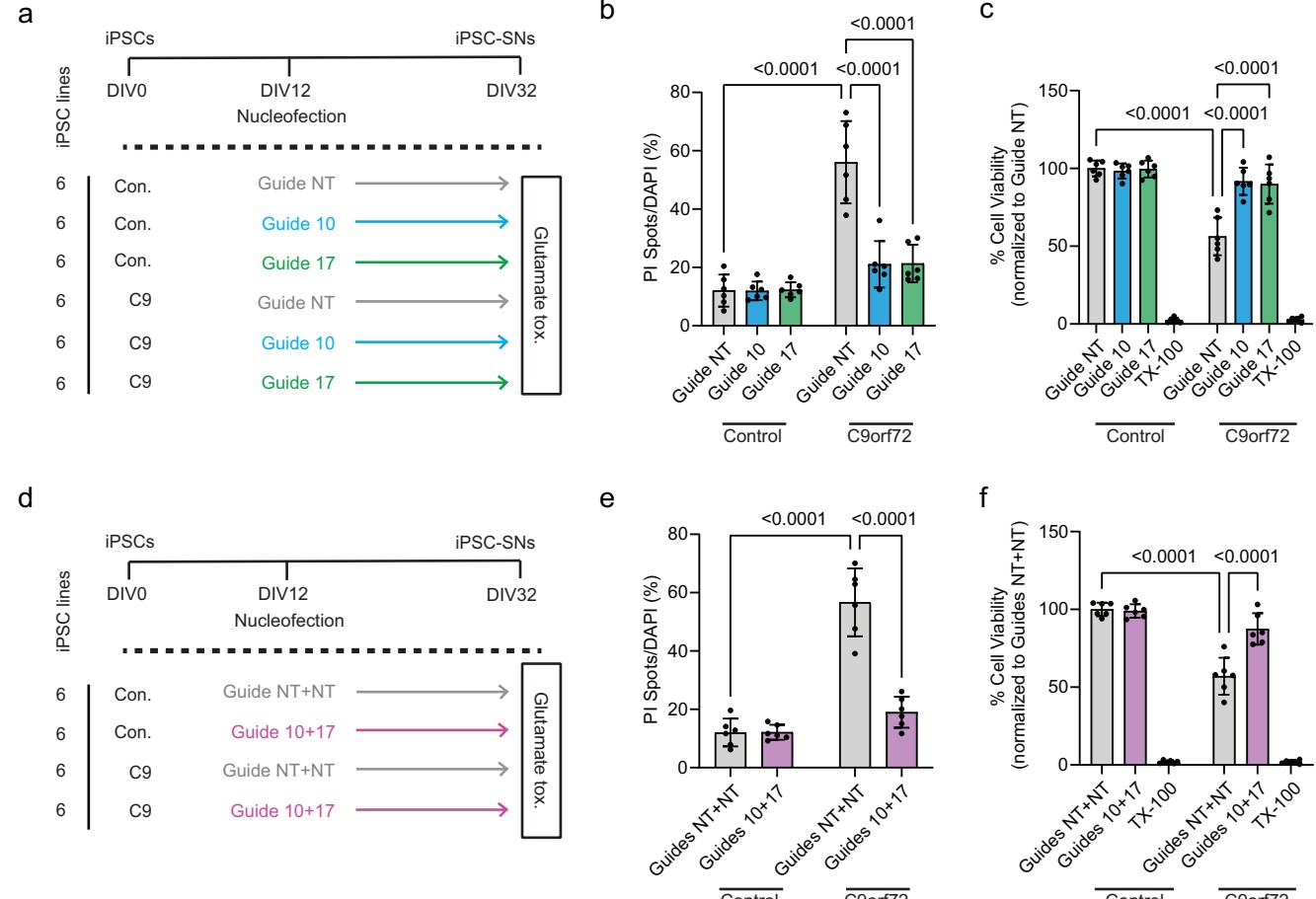

**Fig. 3 | CRISPR-CasRx treatment rescues stressor-induced neurotoxicity in *C9orf72* patient iPSC-derived spinal neurons (iPSC-SNs). a** Glutamate-induced toxicity was assessed 20 days after nucleofection with CRISPR-CasRx plasmids. **b, c** Cell death in control and *C9orf72* iPSC-SNs expressing CRISPR-CasRx plasmids and single gRNAs, quantified as the ratio of propidium iodide (PI) positive spots to DAPI positive nuclei and Alamar Blue cell viability assay following 4-h exposure to 10 μM glutamate. n = 6 lines per condition. **d** Schematic of glutamate-induced excitotoxicity assay in control and C9 lines expressing CRISPR-CasRx dual guide plasmids. **e, f** Quantification of PI incorporation and Alamar Blue cell viability following 4-h exposure to 10 μM glutamate, demonstrating rescue by the dual guide CRISPR-CasRx. Data points for PI incorporation represent average percent cell death across 10 images per well. Data points for Alamar Blue assay represent average percent viability from 3 replicate wells for each condition. Two-way ANOVA with Tukey's multiple comparison test was used to calculate statistical significance. Data presented as mean ± S.D. Source data are provided as a Source Data file.

Fig. S3a) and found that, for our most efficacious guide 17, CasRx levels were highest in C9 line 3, the line with the greatest reduction of antisense repeat transcripts. Finally, we measured the levels of the DPRs polyGP and polyGA using quantitative MSD immunoassays and observed a ~60% reduction in both polyGP (Fig. 2f) and polyGA (Fig. 2g) in all 3 lines, with both guides achieving similar efficacy across all C9 lines. We are unable to detect DPRs specific to the antisense strand in these cells using our current immunoassays so it was not possible to elucidate the level of reduction in antisense DPRs. Taken together, these results show CasRx can achieve robust reduction in endogenous sense and antisense repeat RNAs across multiple patient iPSC-neuron lines.

## *C9orf72* repeat targeting CRISPR-CasRx does not cause toxicity in patient iPSC-neurons

There is clear evidence that activated CRISPR-Cas13 enzymes can cause toxicity due to collateral RNA cleavage when targeting highly expressed transcripts but does not have detectable off-target transcriptional changes or consequent toxicity when targeting low expression transcripts[36]. To determine if there was overt toxicity or off-target transcriptional changes in patient iPSC-neurons when targeting *C9orf72* with CRISPR-CasRx, we performed cell viability assays in two of

our C9 patient lines (C9 lines 1 and 2) 5 days post-transduction. We observed no significant reduction in viable cells after a 5-day treatment with CasRx plus gRNA expressing lentiviruses (Supplementary Fig. S3b, c), though there was a downward trend in C9 line 2 (Supplementary Fig. S3c).

Using the same experimental paradigm, we performed RNA sequencing of three independent inductions of C9 line 1 iPSC-neurons 5 days post-transduction with lentiviruses expressing CasRx and either non-targeting guide, sense transcript targeting guides 8 or 10 (Fig. 2h and Supplementary Fig. S3d), or antisense transcript targeting guides 11 or 17 (Supplementary Fig. S3e, f). For analysis of target engagement of the sense *C9orf72* transcripts, we grouped the *C9orf72* transcripts into two groups as with our RT-qPCR dataset: those transcripts that contain intron 1 and the $G_4C_2$ repeat sequence (variants 1 and 3), and those that do not contain the repeat sequence (variant 2). Guides 8 and 10 significantly reduced repeat-containing *C9orf72* transcripts but not *C9orf72* transcripts that do not contain the repeat expansion (Fig. 2h and Supplementary Fig. S3d). No other differentially expressed genes (DEGs) were observed, indicating no significant off-target transcriptional changes were detectable when using a sequencing depth of 20 million reads per sample. We were not able to detect the antisense repeat-containing transcripts in our RNA-sequencing dataset, likely

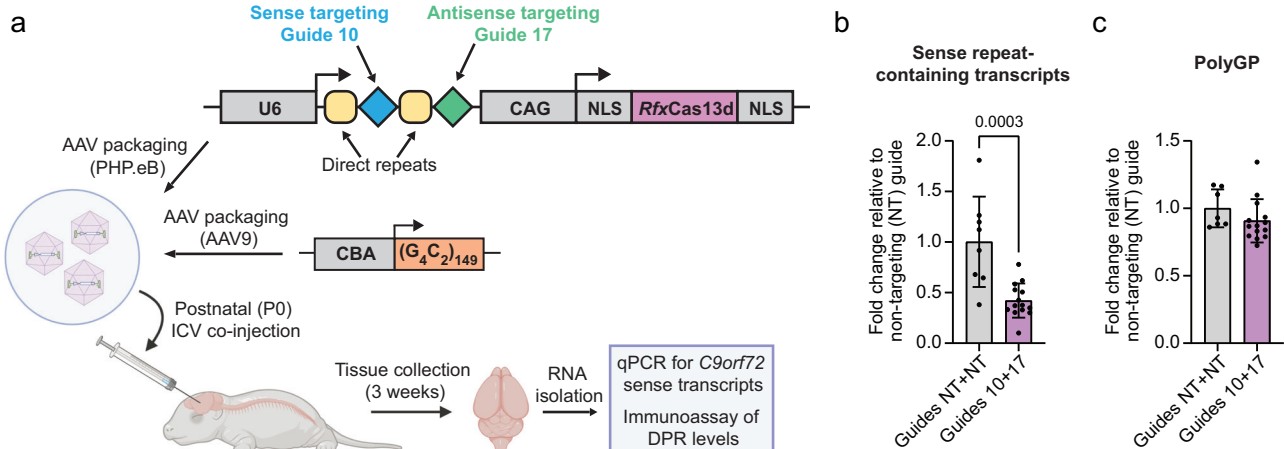

**Fig. 4 | CRISPR-CasRx reduces $G_4C_2$ sense RNA in vivo in 149 R mice. a** Schematic of experimental paradigm. CasRx PHP.eB AAV and the 149 R *C9orf72* AAV9 were co-injected via intracerebroventricular (ICV) injection into P0 C57BL6/J mice. CAG-CMV enhancer/chicken β-actin promoter. CBA chicken β-actin promoter, NLS nuclear localisation signal. Created in BioRender. Cammack, A. (2024) https://BioRender.com/u85i634. **b** RT-qPCR of the sense repeat-containing transcripts from the 149 R AAV performed on RNA isolated from the hippocampus 3 weeks post-injection. Each data point represents one animal. RT-qPCR data presented as fold change compared to non-targeting (NT+NT) CasRx PHP.eB AAV-treated mice. n = 8 NT+NT, n = 14 10+17. **c** MSD immunoassay of polyGP in bulk hippocampal tissue 3 weeks post-injection. n = 7 NT+NT, n = 14 10+17. Data in (**b**, **c**) are shown as mean ± S.D, two-sided unpaired t-test. Source data are provided as a Source Data file.

due to the relatively low number of transcripts and so were not able to confirm target engagement using RNA-seq (Supplementary Fig. S3e, f). However, our more sensitive RT-qPCR for the repeat-containing antisense transcripts did show significant reduction when targeted with CRISPR-CasRx and guide 17 (Fig. 2d). We did not observe any DEGs with guide 11 or guide 17 (Supplementary Fig. S3e, f), indicating that they did not lead to detectable off-target transcriptional changes when using our standard sequencing depth. These data show that after 5 days of treatment in patient iPSC-neurons, *C9orf72* repeat-targeting CRISPR-CasRx does not cause overt toxicity.

## CRISPR-CasRx rescues stressor-induced neurotoxicity and cryptic exon expression in *C9orf72* patient iPSC-derived spinal neurons

We next tested whether CRISPR-CasRx-mediated lowering of C9 sense and/or antisense transcripts could prevent neurotoxic phenotypes and TDP-43 loss of function-associated splicing alterations, in iPSC-spinal neurons (iPSC-SNs), either individually or in dual-targeting configuration (Supplementary Fig. S4a–c). We used a nucleofection paradigm in 6 different *C9orf72* patient iPSC-SNs, which resulted in ~80% reduction of sense or antisense RNAs without loss of variant 2 when treated with CRISPR-CasRx and either guide 10 or 17, respectively, in single vectors (Supplementary Fig. S4d, e). After a 20-day incubation period, we then tested whether sense or antisense knockdown could prevent glutamate-induced excitotoxicity, a well-studied phenotype associated with *C9orf72* expression in iPSC-neurons[37–40], and expression of cryptic exons previously observed in ALS[41–45], which have been previously shown in these lines by this timepoint[20]. We observed that either guide 10 or guide 17 could significantly reduce cell death upon glutamate exposure (Fig. 3a–c), while guide 17 could prevent the inclusion of cryptic exons in *STMN2* and *HDGFL2* (Supplementary Fig. S4g–i). Importantly, neither guide 10 nor 17 had a significant impact on neuronal survival in control iPSC-SNs (Fig. 3a–c) suggesting they are not overtly toxic themselves. We then repeated this assay, but with single vectors expressing both guides 10 and 17 simultaneously. As with the guides by themselves, we observed robust knockdown of sense and antisense RNAs (Supplementary Fig. S4j, k), no reduction in variant 2 (Supplementary Fig. S4l), significant protection from excitotoxicity (Fig. 3d–f), and reduced cryptic exon expression (Supplementary Fig. S4m–o). Thus, CRISPR-CasRx can effectively rescue

*C9orf72* neurodegeneration-associated phenotypes in patient iPSC-derived neurons in vitro.

## CRISPR-CasRx rescues larval hyperactivity in a *C9orf72* repeat zebrafish model

Having established efficacy in vitro, we tested CRISPR-CasRx in vivo using *Tg(ubi:G4C2x45)* zebrafish larvae (Supplementary Fig. S5a), which express 45 pure $G_4C_2$ repeats and generate DPRs[46]. Using an MSD immunoassay, we confirmed that $G_4C_2$ larvae produce polyGP (Supplementary Fig. S5b). To determine whether the model has a larval phenotype, we video-tracked two clutches of $G_4C_2$ larvae over multiple day-night cycles and assessed activity with our previously established automated pipeline[47,48]. Compared to wild-type siblings, $G_4C_2$ larvae were markedly hyperactive (Supplementary Fig. S5c), spending on average 1.7× more time performing swimming bouts each day (Supplementary Fig. S5d). As this model does not contain any of the endogenous *C9orf72* human sequence upstream or downstream of the repeats, we designed new gRNAs specifically to target the zebrafish $G_4C_2$ transcripts (Supplementary Fig. S5e). We injected single-cell embryos with plasmids encoding CasRx and $G_4C_2$-targeting or control non-targeting gRNAs, together with Tol2 recombinase mRNA to integrate the construct in the genome. We screened larvae for high expression of the construct (Supplementary Fig. S5f) and video-tracked them (Supplementary Fig. S5h). As expected, control-injected $G_4C_2$ larvae were hyperactive, 1.3× more time active each day compared to wild-type larvae expressing CasRx and non-targeting gRNAs (Supplementary Fig. S5i). In contrast, expression of CasRx and targeting gRNAs reduced the amount of polyGP DPRs (Supplementary Fig. S5g) and hyperactivity (Supplementary Fig. S5i). CRISPR-CasRx can therefore rescue a behavioural phenotype of an animal model of pathogenic *C9orf72* repeats.

## CRISPR-CasRx effectively reduces sense repeat RNA in $(G_4C_2)_{149}$ repeat mice

We next wanted to determine whether we could use CasRx to target sense repeats in a mouse model. We produced two PHP.eB serotype AAVs expressing CasRx and either our targeting dual gRNAs (guides 10 and 17), or non-targeting gRNAs as a control. We first used a mouse model that utilises neonatal intracerebroventricular (ICV) injection of an AAV expressing 149 pure $G_4C_2$ repeats (149R)[49]. These mice have

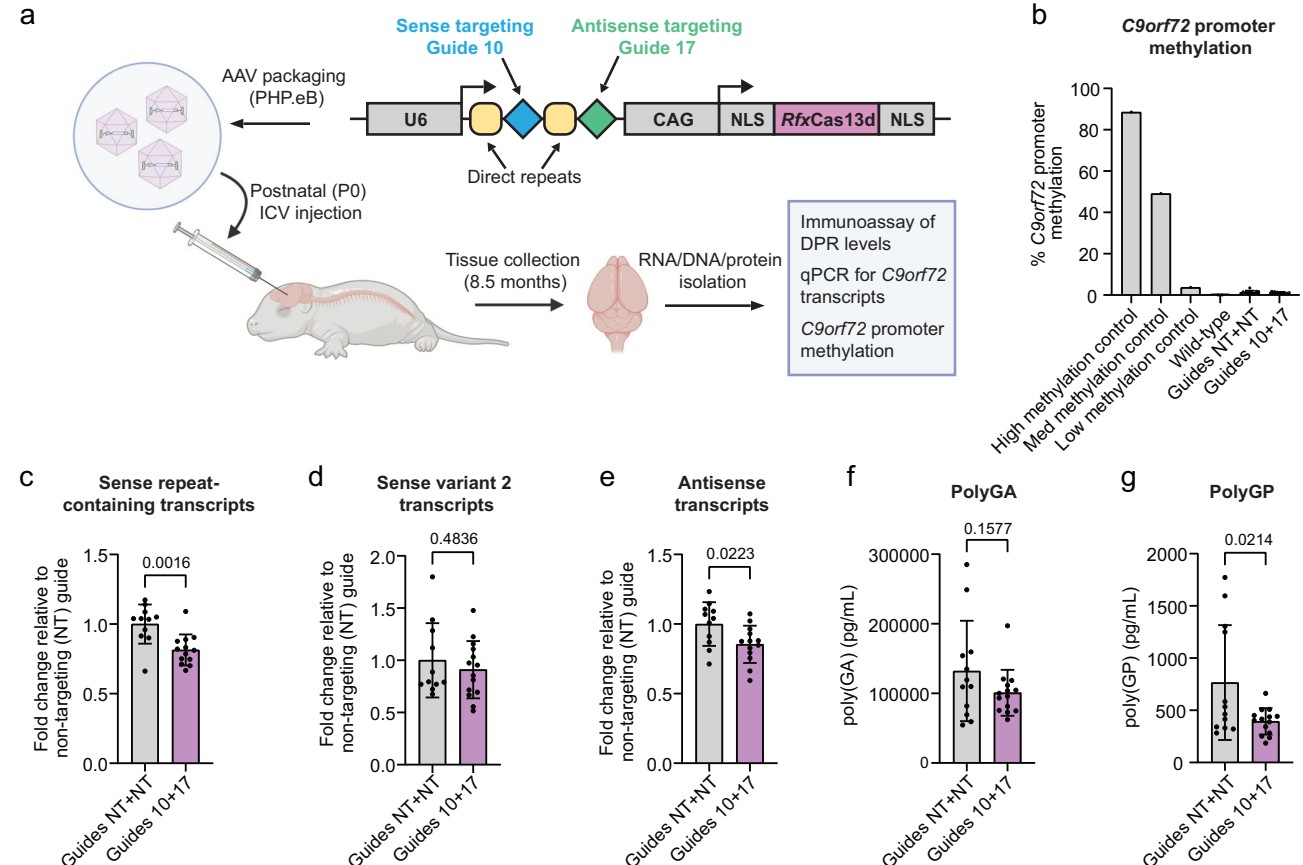

**Fig. 5 | CRISPR-CasRx simultaneously reduces sense and antisense repeat RNAs in *C9orf72* BAC transgenic mice. a** Diagram of experimental paradigm. CasRx PHP.eB AAV (8E+9 vg per animal) was injected via intracerebroventricular (ICV) injection into postnatal day 0 (P0) *C9orf72* BAC mice. Tissue was harvested 8.5 months post-injection. CAG-CMV enhancer/chicken β-actin promoter, NLS nuclear localisation signal. Created in BioRender. Cammack, A. (2024) https://BioRender.com/u85i634. **b** *C9orf72* promoter methylation assay of prefrontal cortical DNA extracted from *C9orf72* BAC mice, demonstrating no overt transgene silencing within the cohort. **c**–**e** RT-qPCRs for *C9orf72* transcripts in *C9orf72* BAC mice treated with CRISPR-CasRx + gRNA AAVs in bulk prefrontal cortex 8.5 months post-injection. RT-qPCR for **c** sense repeat-containing *C9orf72* transcripts (variants 1 and 3). **d** sense variant 2 *C9orf72* transcripts, or **e** antisense repeat-containing *C9orf72* transcripts. RT-qPCR data presented as fold change compared to non-targeting (NT+NT) gRNA CasRx PHP.eB AAV-treated mice. For (**c**–**e**), n = 13 Guide 10+17; n = 11 Guide NT+NT. **e**, **f** Levels of **f** polyGA and **g** polyGP DPRs in bulk cortical tissue from 8.5 month old animals treated with CasRx PHP.eB AAV. For (**e**, **f**), n = 14 Guide 10+17; n = 12 Guide NT+NT. Data in (**c**–**g**) are shown as mean ± SD, two-sided unpaired t-test. Source data are provided as a Source Data file.

been previously shown to produce pathological hallmarks of C9 ALS/FTD including repeat-containing transcripts and DPRs[49]. Additionally, this model contains 119 nucleotides of the endogenous *C9orf72* sequence upstream and 100 nucleotides downstream of the repeats, which means our sense-targeting gRNA (guide 10) will target the sense repeat-containing transcript in this model. However, as our antisense-targeting gRNA (guide 17) targets further from the repeats than the sequence included in this model, it cannot be used to assess our antisense guide. Postnatal day 0 (P0) mice were co-injected via ICV administration with the $(G_4C_2)_{149}$ AAV and either our targeting or non-targeting CasRx PHP.eB AAVs (Fig. 4a). After 3 weeks we assessed levels of the sense repeat-containing transcripts in the hippocampus via RT-qPCR. This revealed a >50% reduction in sense repeat-containing transcripts, confirming successful target engagement in vivo (Fig. 4b). PolyGP levels showed a non-significant reduction at this timepoint, suggesting that longer term expression may be needed for robust DPR reduction in this model (Fig. 4c).

## Dual-targeting CRISPR-CasRx effectively reduces sense and antisense repeat RNA in *C9orf72* BAC mice

We next wanted to determine whether our dual-targeting CRISPR-CasRx AAV could reduce both the sense and antisense repeat-containing *C9orf72* transcripts simultaneously in vivo. We therefore used a bacterial artificial chromosome (BAC) mouse model that contains the full human *C9orf72* sequence and 500 $G_4C_2$ repeats and thus better emulates the human sense and antisense transcripts[50]. *C9orf72* BAC transgenic mice were injected at P0 with a CRISPR-CasRx AAV expressing either our sense and antisense targeting gRNAs in an array, or two non-targeting control gRNAs in an array as a control (Fig. 5a). We chose P0 ICV injection for delivery and packaged our AAVs into PHP.eB capsids to provide the highest possible brain transduction efficiency while avoiding transduction in peripheral tissues, as this serotype shows almost no transduction of the liver after ICV injection[51].

To assess overt toxicity due to AAV expression, we measured weights and conducted a general health and welfare assessment[52,53] in ICV-injected animals monthly starting at 2 months of age and then harvested brain tissue at 8.5 months. In either wild-type or BAC transgenics, sense and antisense targeting CasRx AAVs did not significantly alter weight (Supplementary Fig. S6a, b), induce overt symptoms of sickness (Supplementary Fig. S6c, d), or cause evident microgliosis or astrocyte activation (Supplementary Fig. S6e–l). Thus, CasRx AAV delivery was well tolerated. *C9orf72* BAC transgenics have been described to occasionally develop promoter methylation and partial transgene silencing[54]. Therefore, to ensure that any observed change in *C9orf72* RNA or DPR pathology is not due to promoter silencing, we conducted bisulfite pyrosequencing as previously

described[54] and assessed *C9orf72* promoter DNA methylation levels in the prefrontal cortices of all animals in the study. No animals in the study had a methylated *C9orf72* promoter (Fig. 5b), confirming that any changes in RNA or DPR pathologies could be attributed to CasRx efficacy, rather than transgene silencing. We then assessed bulk prefrontal cortical tissue for target engagement by RT-qPCR and DPR immunoassay. We observed a significant ~20% decrease in both sense and antisense repeat-containing transcripts whilst variant 2 was spared (Fig. 5c–e). There was also no change in C9orf72 protein levels via Western blot (Supplementary Fig. S6m, n), although we note that this band contains both mouse (which is not targeted by CasRx) and human C9orf72. DPR levels were more variable, but were also reduced, with a significant reduction in polyGP (Fig. 5f, g), however surprisingly we observed no change in RNA foci load, assessed in the retrosplenial cortex (Supplementary Fig. S7).

Given the relatively modest lowering of repeat RNAs and DPRs compared to our in vitro assays, we wondered whether this might be due to a lower transduction efficiency in vivo, as our CRISPR-CasRx AAVs were delivered with a relatively low titre (8E+9 vg per animal). As anti-CasRx antibodies are currently unavailable, to address this question we analysed the neuronal tropism and transduction of an identical PHP.eB AAV wherein we replaced the CasRx sequence with a 2xNLS-mApple fluorescent reporter allowing for expression under the exact same promoter and regulatory sequences as our CRISPR-CasRx vectors. We then delivered this AAV to P0 BAC mice with the same titre and delivery method as CasRx and measured mApple expression across the brain 4 weeks later. We observed that transduction efficiency in this paradigm is only ~5–10% of all cells across the cortex and hippocampus and that transduction is largely confined to neurons and astrocytes (Supplementary Fig. S8). This suggests that in the cells which received CasRx treatment, there was a robust knockdown of repeat RNAs, but that this is diluted in bulk tissue due to the low transduction efficiency, which explains the more modest effects observed in vivo. Taken together these data show that our approach can simultaneously reduce both sense and antisense repeat transcripts in vivo.

## Discussion

Here we detail a CRISPR-CasRx paradigm for lowering sense and antisense *C9orf72* repeat transcripts and demonstrate efficacy in transiently expressing cell models, patient iPSC-derived neurons, and *C9orf72* mice without affecting variant 2 expression. A key aspect of CasRx is its ability to mature its own gRNA array, allowing for multiplexed targeting of multiple transcripts at one time, while its small size allows for packaging into a single AAV vector. In this study, we utilise this feature to simultaneously target and reduce sense and antisense repeat transcripts, two important pathologies thought to drive neurotoxicity in *C9orf72* ALS/FTD[22]. In addition, while a large body of evidence suggests that toxic gain of function is a major driver of *C9orf72*-related ALS/FTD[4], loss or reduction of *C9orf72* has also been shown to exacerbate gain of function toxicity[55,56]. Therefore, an important part of any therapy should be to minimise any further reduction in *C9orf72* expression. To this end, we targeted our sense guides to the sequence upstream of the repeat expansion in intron 1, thus targeting only those transcripts that contain the repeats. This strategy successfully reduced endogenous repeat-containing transcript variants 1 and 3, whilst sparing variant 2 in all models tested here. Importantly, we observe here that *C9orf72* targeting with CasRx was able to prevent established neurodegenerative phenotypes in vitro and in vivo. In iPSC-SNs, CasRx treatment prevented glutamate-induced stress and cryptic exon inclusion, while in a *C9orf72* zebrafish model, we observed reduced hyperactivity, a phenotype consistently observed in other in vivo C9 models[49,57,58]. Thus, CasRx treatment provides a functional benefit in protecting against repeat-induced neurotoxicity.

For any genetic therapy it is vital to minimise off-target effects. CasRx leads to collateral cleavage of RNAs when it is targeted to highly-expressed transcripts, with less collateral cleavage when targeting transcripts with lower expression levels[36]. We did not observe off-targets using bulk RNA-sequencing with 20 million reads per sample when treating *C9orf72* patient iPSC-neurons. This is consistent with the original report of CasRx, which showed no significant off-target transcriptomic changes when targeting *B4GALNT1* in mammalian cells, compared to over 500 off-target changes when targeted using a comparable RNA interference method[27]. However, it is unlikely that there are no off-target effects, and more in-depth and sensitive analyses may be required to reveal low-level and unique off-target events.

To date there are no therapies in clinical trial for C9 ALS/FTD that target both the sense and antisense repeat-containing *C9orf72* RNA. This is despite a growing body of evidence suggesting the antisense repeat expansion RNA contributes to disease pathogenesis. This includes the overt toxicity of antisense-derived polyPR in vitro and in vivo[17–19]; recent work showing antisense repeat-containing RNA triggers activation of the PKR/eIF2α-dependent integrated stress response independent of DPRs and sense strand-related pathology, leading to aberrant stress granule formation[21]; and data showing that only ASOs targeting antisense repeats are able to reduce TDP-43 loss of function cryptic splicing changes in *C9orf72* patient iPSC-SNs[20]. Indeed, when we treated *C9orf72* patient iPSC-SNs with CRISPR-CasRx, we found that while both guides 10 and 17 could independently reduce excitotoxicity, only the antisense-targeted guide 17 could prevent cryptic exon inclusion, further demonstrating the importance of antisense pathologies to disease. The need to target antisense repeats appears particularly compelling given the recent failures of two clinical trials targeting sense repeats, which did not show clinical efficacy despite reducing sense pathologies (https://investors.biogen.com/news-releases/news-release-details/biogen-and-ionis-announce-topline-phase-1-study-results; https://www.thepharmaletter.com/article/wave-life-sciences-ends-wve-004-program)[22,23]. Our data show that CasRx is one plausible approach for targeting both sense and antisense repeat transcripts.

It has previously been shown that some people already possess antibodies to certain Cas9 orthologs[59], although the immune response does not trigger extensive cell damage in vivo[60]. One potential limitation to our current approach is the necessity for long-term expression of CasRx. A recent study identified CasRx-reactive antibodies, and CD4 and CD8 T cell responses in human samples[61]. This was surprising as CasRx is isolated from *Ruminococcus flavefaciens* strain XPD3002 which is bovine-specific and is not present in humans. However CasRx shares ~35% homology with another Cas13 ortholog isolated from *R. bicirculans* that does colonise humans[62]. Importantly, there is a precedent that T-cell responses are manageable with immune suppression, suggesting these issues are surmountable[63].

In concordance with this being a proof-of-concept study, there were some limitations of note in our two mouse studies. First, while P0 ICV is an effective method for delivering high loads of AAV directly to the brain and contributed to the observed efficacy of CasRx despite the low titre used in this study, it will be important in future studies to trial more therapeutically relevant delivery methods, such as intrathecal injection, and at adult timepoints. Second, in a co-AAV delivery paradigm with the 149R model, we observed a large (>50%) reduction in sense repeat-containing RNA, however surprisingly polyGP remained unchanged. It's possible that our 3-week study period was not enough time to reduce polyGP due to its longer half-life or could suggest that a substantial amount of polyGP is deriving from the antisense strand in this model, which we did not target. Similarly, despite a ~20% reduction in repeat RNA in the BAC mice, we observed no change in sense RNA foci and only modest reductions in DPRs, though variability between animals was high in measures of both pathologies, perhaps limiting our ability to observe small changes. In addition, using an mApple reporter AAV matched in all technical aspects to our CasRx AAVs (e.g. titre, delivery method, vector,

etc.), we observed a low (-5–10%) transduction efficiency in the brain, which likely accounts for the modest efficacy of CasRx in BAC transgenic mice compared to our in vitro studies. Delivery of gene therapies to the CNS remains one of the greatest challenges in the field. The significant reduction we observed in sense and antisense RNA despite the low transduction efficiency indicates that there were high levels of knockdown in the cells that were successfully transduced. Thus, delivery with an appropriate capsid and route of administration to enable widespread transduction should provide successful targeting. In summary, our proof-of-concept data suggest CRISPR-CasRx and our gRNAs have potential therapeutic utility via targeting both the sense and antisense repeat-containing *C9orf72* transcripts in a single design small enough to be packaged into an AAV.

## Methods

### Ethics statement

*C9orf72* patient iPSC lines for FACS experiments were kindly provided by Prof Siddharthan Chandran and Dr Bhuvaneish Selvaraj, University of Edinburgh and have been previously described[37] (see Supplementary Table 1 for demographics); they were collected with prior informed patient consent and derived from biopsied fibroblasts. Ethical approval was received from the NHS Health Research Authority East of England—Essex Research Ethics Committee (REC reference 18/EE/0293). Animals were maintained and experimental procedures performed in accordance with the UK Animal Scientific Procedure Act 1986, under project and personal licenses issued by the UK Home Office, and approved by the UCL and the Institute of Prion Diseases Animal Welfare and Ethical Review Bodies. Control and *C9orf72* iPSCs used for excitotoxicity and cryptic exon inclusion assays were obtained from the Answer ALS repository at Cedars Sinai (see Supplementary Table 2 for demographics).

### Plasmid construction

NanoLuc reporter plasmids: the sense NLuc reporter was as previously published[33]. To generate an antisense RAN translation NLuc reporter construct, 680 nucleotides of the endogenous *C9orf72* sequence upstream of the antisense repeats were synthesised and cloned into the NLuc backbone expression vector and the $G_4C_2$ repeats from the sense NLuc plasmid were then cloned in the reverse direction using NotI and BspQI to make $C_4G_2$ repeats with the endogenous upstream sequence. A repeat insert of -55 $C_4G_2$ repeats was confirmed by sizing on an agarose gel. gRNA plasmids: guides were designed using BLAST to identify off-targets and secondary structure scores were predicted using RNAfold Webserver (University of Vienna). We used a previously published non-targeting control guide[26]. Annealed guide oligonucleotides were designed to have overhangs for cloning into their respective gRNA expression backbone vectors using BbsI. gRNA-CasRx lentiviral plasmids: the CasRx lentiviral vector (pXR001) was used as a backbone and primers were designed to PCR out the U6 promoter, direct repeats, and gRNA from the guide expressing vector (pXR003) with PacI restriction site overhangs to allow for cloning into the CasRx backbone and confirmed by sequencing. gRNA sequences are listed in Supplementary Table 3. CasRx and 2xNLS-mApple AAV-PHP.eB plasmids: an insert containing our CRISPR pre-gRNA array and CasRx driven by a 643 bp CAG variant promoter[64] was synthesised (GeneArt, Thermo Fisher) and was cloned into an AAV backbone vector using restriction sites NotI and AscI. Golden Gate cloning utilising type IIs restriction enzyme BsmBI was then used to clone in our guide array of choice (guides 10+17). Correct insertion of the CasRx insert and gRNA array as well integrity of the ITRs was confirmed via Sanger sequencing. To create the 2xNLS-mApple reporter constructs, we PCR amplified and subcloned a 2xNLS-mApple reporter from a previous construct[40] in place of the CasRx sequence in our CRISPR-CasRx vector with NT+NT guides. In brief, the CaxRx/NT-NT vector was digested with ApaI and BsaI (New England Biolabs) and the vector band was gel purified.

The 2xNLS-mApple sequence was amplified with PCR using primers with homology-overhangs and inserted into the AAV backbone with In-Fusion cloning (Takara). Final clone was confirmed with diagnostic restriction digests (including SmaI for ITR integrity) and Sanger sequencing, including through the ITR sequences.

### HEK293T cell culture

HEK293T cells (ATCC) were maintained in DMEM media supplemented with 10% FBS, 4.5 g/L glucose, 110 mg/L sodium pyruvate, 1× GlutaMAX and kept at 37 °C with 5% $CO_2$ to ensure physiological temperature and pH. Cells were maintained up to a confluency of 90% and then dissociated and passaged with 0.05% Trypsin-EDTA.

### Lentivirus and AAV production

Low passage HEK293T cells were cultured as described above and plated in T175 flasks at 50% confluency 24 h prior to transfection with 14.1 µg of PAX lentiviral packaging vector, 9.36 µg of VSV.G lentiviral enveloping vector and 14.1 µg of lentivirus plasmid of interest using Lipofectamine 3000. Twenty-four hours after transfection, cell media was replaced with NGN2 neuronal induction media. The media was collected 24 h later and filtered to remove cell debris via a 45 µm filter before being aliquoted and stored at −80 °C until needed. To produce PHP.eB serotype AAVs, low passage HEK293T cells were triple transfected with plasmids containing our expression cassette flanked by 5′ and 3′ inverted terminal repeats (ITRs), a PHP.eB Cap/Rep plasmid, and a helper plasmid in a 1:1:1 plasmid copy number ratio. Seventy two hours post-transfection supernatant and cells were collected, and virus was purified via ultracentrifugation in an increasing gradient of iodixanol at $200,000 \times g$ for 3 h with isolation of viral particles in the 40% fraction. This fraction was then filtered, concentrated via Vivaspin™ 20 concentrator (100.000 MW cutoff) and titered by qPCR. PHP.eB CasRx AAVs were made at the GeneTxNeuro Vector Core Facility at the UCL School of Pharmacy and a second batch was outsourced to the viral vector facility at ETH Zurich. 149 R AAV9 viral particles were generated as previously described[49]. 2xNLS-mApple AAVs were generated in-house.

### NanoLuciferase reporter assays

For luciferase assays, HEK293T was plated at a density of 30,000 cells per well (96 well plate) and transiently transfected with 100 ng of Cas13 gRNA plasmids, 25 ng of Cas13 plasmid, 12.5 ng of firefly luciferase expression plasmid, and 2.5 ng of RAN translation sense or antisense NanoLuciferase reporter plasmids (S92R-NL and AS55R-NL respectively). Transfection reagents were added directly to the media and left on for the duration of the experiment. Each experiment consists of 3–5 technical replicate wells per condition. Forty eight hours post-transfection both firefly and NanoLuciferase signals were measured using the Nano-Glo Dual Luciferase Assay (Promega) according to manufacturer's instructions, on the FLUOstar Omega (BMG Labtech) with a threshold of 80% and a gain of 2000 for both readings. The NanoLuciferase reading was normalised to the firefly luciferase reading for each well to control for variable transfection efficiencies.

### Combined single molecule RNA FISH and immunocytochemistry

For RNA fluorescent in situ hybridisation (RNA FISH) in HEK293T cells, cells were plated at 25,000 cells per well in a 96-well plate and transfected with 100 ng of CasRx gRNA plasmids, 25 ng of CasRx plasmids, 12.5 ng of firefly luciferase expression plasmid, and 2.5 ng of RAN translation sense or antisense NanoLuciferase reporter plasmids. Each plate contained 3–5 technical replicates per condition. Cells were fixed 48 h post-transfection for 7 min using 4% paraformaldehyde (PFA) with 10% methanol, diluted in PBS. Cells were then dehydrated via 70% and 100% ethanol washes and frozen at −80 °C in 100% ethanol until needed. Cells were rehydrated with 70% ethanol and washed for 5 min at

room temperature in pre-hybridisation solution (40% formamide, 2× SSC, 10% dextran sulphate, 2 mM vanadyl ribonucleoside complex). Cells were permeabilised with 0.2% Triton X-100 for 10 min. Cells were incubated at 60 °C in pre-hybridisation solution for 45 min. Locked nucleic acid (LNA) probes (Qiagen) to detect either sense (5′ TYE563-labelled CCCCGGCCCCGGCCCC) or antisense (5′ TYE563-labelled GGGGCCGGGGCCGGGG) RNA-foci were then added to the pre-hybridisation solution at 40 nM and cells were kept in the dark at 60 °C or 66 °C (for sense or antisense probes, respectively) for 3 h or overnight. Cells were then washed with 0.2% Triton-X100 in 2x SSC for 5 min at room temperature followed by 30 min at 60 °C. One additional wash in 0.2x SSC at 60 °C for 30 min was performed prior to application of 647-conjugated HA antibody (in 0.2x SSC with 1% BSA at 1:1000) for detection of HA-tagged CasRx. This was incubated overnight at 4 °C and then washed with 0.2x SSC for 20 min at room temperature. Hoechst was added at 1:5000 in 0.2x SSC for 10 min to detect cell nuclei. Hoechst solution was removed, and cells were left in 0.2x SSC at 4 °C protected from light until imaging.

### iPSC culture for FACS experiments
The three *C9orf72* patient-derived iPSC lines used are described in Supplementary Table 1. iPSCs were cultured on Geltrex-coated wells, fed with fresh E8 media (Gibco) daily, and passaged with EDTA (0.5 mM).

### Generation of NGN2-inducible iPSC neurons
*C9orf72* patient-derived iPSCs expressing doxycycline-inducible NGN2 iPSCs were generated by piggyBac integration as previously described[34,35]. Cells underwent FACS using the BFP in the NGN2 construct to ensure a pure population of NGN2 iPSCs. To avoid integration artefacts due to piggyBac random insertion, sorted NGN2 iPSCs were kept as a mixed population and used at low passage numbers.

### Differentiation of NGN2-inducible iPSCs into cortical-like neurons
iPSCs were differentiated to cortical-like neurons via dox-inducible NGN2 expression as previously described[65]. Briefly, the cells were cultured to ~80% confluency prior to differentiation. On DIV0 of neuronal differentiation, NGN2 iPSCs were dissociated and lifted with Accutase (Gibco). Cells were replated at desired ratio ($3.75 \times 10^5$ cells per well/of a 6-well plate) on Geltrex-coated wells in neuronal induction media: DMEM-F12 (Gibco) containing 1x N2 (Thermo Fisher Scientific), 1x Glutamine (Gibco), 1x HEPES (Gibco), 1x NEAA (Gibco), doxycycline (2 μg/mL) and 10 μM Rock inhibitor Y-27632 (DIV0 only; Tocris). Cells were washed with PBS the following day to remove cell debris and fresh induction media was added without Rock inhibitor. On DIV3, induction media was removed and replaced with neuronal maintenance media: Neurobasal (Gibco), supplemented with 1x B27 (Gibco), 10 ng/mL BDNF (PeproTech), 10 ng/mL NT-3 (PreproTech) and 1 μg/mL laminin. Cells were given a half-media change on DIV4 with fresh maintenance media and then used for downstream processing on DIV5 as indicated.

### Immunocytochemistry of iPSC-derived cells
iPSCs were differentiated as described above. On day 3 of the differentiation protocol, neural precursor cells were replated onto laminin-coated 96-well ViewPlates (Revvity) and cultured in maintenance media. At DIV5, cells were washed with 1x PBS (Gibco) before fixation with 4% PFA for 10 min at room temperature, followed by another washing step. Cells were then treated with blocking buffer (4% BSA in PBS with 0.3% Triton-X) for 1 h at room temperature. Primary antibodies were diluted to working concentration (Supplementary Table 4) in 0.3% PBS-Triton-X and added to cultures for over-night incubation at 4 °C. After a wash cycle (3 × 5 min) with washing buffer (0.3% PBS-Triton-X), cells were incubated with secondary antibodies in blocking buffer for 90 min at room temperature in the dark. Following

another wash cycle, Hoechst-staining was performed for 5 min (Invitrogen, 1:10,000 in PBS). Finally, cells were washed 1x with PBS and stored in PBS until imaging with a Zeiss 880 confocal microscope.

### Lentiviral delivery of CasRx and guide RNA to NGN2 iPSC-neuron and fluorescence-activated cell sorting
NGN2 iPSCs were treated with lentiviruses on DIV0 of the differentiation protocol, with lentiviral supernatant directly from the HEK293T cell viral production protocol, 2 h post dissociation with Accutase. Twenty-four hours later, media was replaced with fresh induction media and the differentiation protocol followed. At DIV5, cells were lifted and dissociated with Accutase for downstream analyses. For FACS, cells were pelleted via centrifugation prior to resuspension in FACS buffer (PBS + 1% BSA + 2 mM EDTA). Cell clumps were removed via a 100 μm cell strainer and cells were resuspended in FACS buffer at $3 \times 10^6$ cells/mL. Cells were sorted at the UCL Flow Cytometry Core Facility using the BD FACSAria™ III Cell Sorter. Sorted cells were collected and pelleted via centrifugation and snap-frozen for use in downstream analyses.

### Protein extraction, Meso Scale Discovery (MSD) immunoassays, and western blots
To extract protein from the NGN2-iPSC neurons for immunoassays, FACS purified cell pellets were incubated in lysis buffer (2% SDS + protease inhibitors in RIPA buffer) for 5 min at room temperature prior to sonication for $3 \times 5$ s at 30 amp at 4 °C. Samples then underwent centrifugation at $17,000 \times g$ at 4 °C for 20 min to remove cell debris. To extract protein from *C9orf72* BAC mouse tissue, 100 mg of frozen prefrontal cortex was taken per mouse sample, lysed with 0.9 x tissue mass of lysis buffer (1x RIPA buffer with 2% SDS and 2x protease inhibitor cocktail), and homogenised using a TissueRuptor II (Qiagen). Zebrafish larvae were similarly extracted, with 0.9 x tissue mass of lysis buffer (1x RIPA buffer with 2% SDS and 2x protease inhibitor cocktail) and TissueRuptor II (Qiagen) homogenization. Samples were then sonicated at 4 °C for $3 \times 20$ s at 30% amplitude. Samples were centrifuged at $17,000 \times g$ for 20 min at 16 °C and supernatant was collected. Protein concentration was determined via BCA assay (Thermo Fisher) according to manufacturer's instructions. MSD assays were performed as previously described[66,67]. For polyGP, capture and detection antibody was the same affinity-purified custom anti-(GP)8 antibody (Eurogentec, biotinylated for detector). For polyGA, capture antibody was anti-GA (Millipore, # 5E9-3226534) and detector antibody was monoclonal anti-GA 5F2 kindly gifted by Prof Dieter Edbauer (DZNE Munich) and biotinylated in-house. Western blot for C9orf72 and GAPDH was performed as previously described[68] with minor modifications. In brief, 20 μg of prefrontal cortex protein was loaded per sample, separated on a 4% to 12% NuPAGE bis-tris gel (Invitrogen), and transferred to a nitrocellulose membrane (Bio-Rad) with semi-dry transfer. The membrane was blocked with 5% milk in PBS with 0.1% Tween-20 (Sigma) for 1 h and probed with primary antibodies diluted in 5% milk/PBS-T (C9orf72 GeneTex, GTX634482, 1:1000; Gapdh Cell Signalling, 14C10, 1:1000) overnight at 4 °C. The next day, blot was washed 3 times with PBS-T, incubated with AlexaFluor-conjugated secondary antibodies for 1 h at room temperature, washed again, and visualised with a Bio-Rad ChemiDoc MP Imaging System. Band densities were quantified with ImageJ by calculating area under the curve in densitometry plots and normalised to Gapdh.

### RNA extraction and RT-qPCR
Total RNA was extracted from FACS-purified iPSC-neurons and frozen mouse brain tissue using the ReliaPrep RNA cell kit and ReliaPrep RNA tissue extraction kit (Promega) respectively. RNA samples underwent reverse transcription using SuperScript™ IV VILO™ Master Mix (ThermoFisher) with random hexamers to produce cDNA according to manufacturer's instructions. qPCR analysis was performed using SYBR

Green Master Mix (ThermoFisher) for all qPCRs except *C9orf72* variant 2 expression levels for which TaqMan probes were used (sequences provided in Supplementary Table S5) and measured with a LightCycler™ (Roche). qPCR was performed in technical triplicate with data normalised to GAPDH expression levels.

### RNA sequencing and analysis

RNA library was prepared for sequencing with KAPA RNA HyperPrep with RiboErase. RNA-sequencing (NextSeq 2000) was carried out at a depth of 20 million reads per sample. RNA-seq reads were pseudoaligned to the reference human transcriptome (Ensembl GRCh38 v96) using kallisto (v.0.46.0)[69]. Differential gene expression analysis was performed using the DEseq2 package (v1.36.0)[70] by importing the estimated abundance data using the tximport package[71]. The threshold for differentially expressed genes was an adjusted p-value of less than 0.05 and $\log_2$(fold change) >1. R packages EnhancedVolcano (v.1.14.0), ggplot2 (v.3.4.1), and ggsci (v.2.9) were used for data visualisation. The transcripts of *C9orf72* were categorised into two groups based on the presence or absence of exon 1a or 1b, with the following Ensembl-annotated isoforms: Exon 1a-containing: ENST00000619707.4, ENST00000647196.1, ENST00000379997.7. Exon 1b-containing: ENST00000379995.1, ENST00000380003.7, ENST00000644136.1, ENST00000488117.5

### Cell viability assay

NGN2 iPSCs were plated on Geltrex-coated 96-well microplates and differentiation and transduction protocols were followed as above. Cell viability was assessed 5 days post-transduction via CellTiter-Fluor™ assay according to manufacturer's instructions. 20 mg/mL digitonin (Merck) was added as a positive control. Fluorescent signals (380–400 $nm_{Ex}$/505 $nm_{Em}$) were determined via analysis with the PHERAstar™ microplate reader (BMG Labtech).

### Spinal neuron differentiation, excitotoxicity assays, and cryptic exon measurement

iPSCs were maintained in mTeSR Plus medium at 37 °C with 5% $CO_2$. iPSC-derived spinal neurons (iPSC-SNs) were differentiated according to a modified diMNs protocol[39,72–74] and maintained at 37 °C with 5% $CO_2$. iPSCs and iPSC-SNs were routinely tested negative for mycoplasma. For glutamate excitotoxicity assays, iPSC-SNs were dissociated with trypsin on day 12 of differentiation. $5 \times 10^6$ iPSC-SNs were nucleofected with 4 µg plasmid DNA in suspension as previously described[39,73]. Following nucleofection, 100 µL of cell suspension was plated in each well (total of 6 wells per cuvette) of a glass bottom or plastic 24 well plate for PI and Alamar blue toxicity and viability experiments respectively. Media was exchanged daily for a total of 20 days to facilitate the removal of iPSC-SNs that failed to recover post-nucleofection. On the day of the experiment (day 32 of differentiation), iPSC-SN media was replaced with artificial CSF (ACSF) solution containing 10 µM glutamate. For those iPSC-SNs undergoing Alamar Blue viability assays (plastic dishes), Alamar blue reagent was additionally added to each well according to manufacturer protocol at this time. Following incubation, iPSC-SNs for PI cell death assays were incubated with PI and NucBlue live ready probes for 30 min and subjected to confocal imaging as previously described[39]. The number of PI spots and nuclei were automatically counted in FIJI. Alamar Blue cell viability plates were processed according to manufacturer protocol. As a positive control, 10% Triton X-100 was added to respective wells 1-h prior to processing. For cryptic exon inclusion assays, iPSC-SNs were dissociated with trypsin on day 12 of differentiation. $5 \times 10^6$ iPSC-SNs were nucleofected with 4 µg plasmid DNA in suspension as previously described[39,73]. Following nucleofection, 600 µL of cell suspension was plated in each well of a 6-well plastic plate (total of 1 well per cuvette). Media was exchanged every other day for 20 days. RNA isolation was carried out using phenol-chloroform extraction and cDNA synthesised using the High Capacity cDNA synthesis kit (Thermo Fisher) as

previously described[20,73,74]. 1 µg RNA was used for all cDNA reactions. qRT-PCR was carried out with SYBR Green Master Mix and an Applied Biosystems QuantStudio 3 (Applied Biosystems). Primers were synthesized by IDT and sequences are as described in Supplementary Table 5. GAPDH was used as a reference gene.

### Zebrafish experiments

Adult zebrafish were reared by University College London's Fish Facility on a 14 h:10 h light:dark cycle. All experiments used the progeny of an outcross of heterozygous *Tg(ubi:G4C2x45; hsp70:DsRed)*[46] to wild type (AB × Tup LF). To obtain eggs, pairs of one female and one male were isolated in breeding boxes overnight, separated by a divider. Around 9 AM (lights on) the next day, the dividers were removed, and eggs were collected 7–10 min later. The embryos were then raised in 10-cm Petri dishes filled with fish water (0.3 g/L Instant Ocean) in a 28.5 °C incubator on a 14 h:10 h light:dark cycle. Debris and dead or dysmorphic embryos were removed every other day with a Pasteur pipette under a bright-field microscope and the fish water replaced. At the end of the experiments, larvae were euthanised with an overdose of 2-phenoxyethanol (ACROS Organics). Experimental procedures were in accordance with the Animals (Scientific Procedures) Act 1986 under Home Office project licences PA8D4D0E5 and PP6325955 awarded to Jason Rihel. Adult zebrafish were kept according to FELASA guidelines[75].

In the experiment in Supplementary Fig. S5c, d, each clutch was from a unique pair of parents. In each CasRx experiment (Supplementary Fig. S5e–i), single-cell embryos from different pairs were pooled and injected in the yolk with 10 pg of CasRx targeting or non-targeting plasmid and 500 pg of Tol2 mRNA in 1 nL. Only larvae with strong GFP expression were kept for the behaviour tracking experiment. At 5 dpf, individual larvae were transferred to the wells of clear 96-square well plates (Whatman). To avoid any potential localisation bias during the tracking, wild-type and *ubi:G4C2x45* heterozygous larvae (identified by DsRed expression) were plated in alternating columns of the 96-well plate. The plates were placed in ZebraBoxes (Viewpoint Behaviour Technology). From each well, the video-tracking software (ZebraLab, Viewpoint Behaviour Technology) recorded the number of pixels that changed intensity between successive frames. To be counted, a pixel must have changed grey value above a sensitivity threshold, which was set at 20. The metric, termed Δ pixel, describes each animal's behaviour over time as a sequence of zeros and positive values, denoting if the larva was still or moving. Tracking was performed at 25 frames per second on a 14 h:10 h light:dark cycle for ~65 h, generating sequences of roughly 5,850,000 Δ pixel values per animal. In plots (Supplementary Fig. S5c, h), the first day and night (5 dpf) are removed as acclimatisation period. The day light level was calibrated at 555 lux with a RS PRO RS-92 light meter (RS Components). Night was in complete darkness with infra-red lighting for video recording. Both mornings shortly after 9 AM (lights on), the wells were manually topped-up with fish water to counteract water evaporation. At the end of the tracking, any larva that did not appear healthy was excluded from subsequent analysis. Larvae were snap-frozen in liquid nitrogen for DPR measurements.

Behavioural data analysis was performed using the FramebyFrame R package v0.13.0[76]. In Supplementary Fig. S5d, i, time spent active per day was statistically compared between genotypes using linear mixed effects (LME) modelling implemented in the lmer function of the R package lme4 v1.1.31[77]. The command to create the LME model was:

$$lmer(parameter \sim group + (1|experiment/larva) + (1|dpf))$$

Each group was then compared to the reference group using estimated marginal means implemented in the R package emmeans, which provided the p-value.

For the pictures in Supplementary Fig. S5a, f, larvae were anaesthetised and mounted in 1% low melting point agarose (Sigma-Aldrich) in fish water. Pictures were then taken with an Olympus MVX10 microscope connected to a computer with the software cellSens (Olympus).

### *C9orf72* 149R and BAC mouse model housing, AAV administration, and tissue harvesting

C57BL6/J mice were used for 149R experiments. Bacterial artificial chromosome (BAC) *C9orf72* mice[50] in the FVB/NJ background were obtained from the Jackson Laboratory (RRID:IMSR_JAX:029099) and were PCR genotyped using the Jackson Laboratory protocol. Animals were housed in 12 h light/dark conditions with tightly regulated temperature and humidity within a UCL Biological Safety Unit. Within 24 h of birth, pups were manually injected with AAVs via intracerebroventricular (ICV) injection into both hemispheres. Briefly, AAVs were diluted in sterile PBS to a final volume of 5 μL per animal. P0 pups were anaesthetized with isoflurane, after which a calibrated Hamilton 10 μL syringe was inserted into the skull ~2/5 of the distance from lambda to the eye and at a depth of ~2 mm. 2.5 μL of AAV/PBS solution was injected slowly into each hemisphere. After injection, pups were allowed to recover on a heat pad and then returned to the dam. 149R AAV was administered at 6E +10 vg per animal as described previously[49], while CasRx PHP.eB AAVs were administered at 8E+9 vg per animal. For tissue collection, animals were anaesthetised with isoflurane and perfused with ice-cold PBS. Brains were immediately collected, dissected, snap-frozen on dry ice, and stored at −80 °C until use. All in vivo experiments used both male and female animals in Mendelian ratios.

### *C9orf72* BAC promoter methylation assay

Promoter methylation for BAC transgenics was assessed via bisulfite pyrosequencing, conducted by EpigenDx (Assay ADS3232-FS1) as previously described[54]. ADS3232-FS1 covers 16 CpG's within the *C9orf72* promoter, spanning the transcription start site, from base pairs −55 to +125.

### Mouse clinical scoring

To assess the general health and welfare of ICV-injected animals post-AAV delivery, we conducted clinical scoring monthly beginning at 2 months of age until the end of the study (8.5 months of age). The composite clinical score (18-point scale) is adapted from similar health assessments[52,53,78] and assesses 6 aspects of general health and fitness: weight (% change from previous timepoint), posture, coat/grooming, eye squinting, activity, and breathing. Each component is scored 0–3 and are added together for a composite score of 0–18. A composite score of 0–3 is considered healthy/no burden of distress, while 4–8 is mild, 9–12 is moderate, and 13–18 is severe requiring culling.

### RNA FISH and immunohistochemistry in mouse brains

For identification of RNA foci in BAC mice, 5 μm-thick brain sections were deparaffinized in xylene and rehydrated through a series of ethanol solutions, followed by antigen retrieval for 15 min in 1x citrate buffer (420902, BioLegend) at 95 °C. Sections were then dehydrated through a series of ethanol washes and air-dried. After washing in 2x SSC 0.1% tween, slides were incubated in HCR™ (Molecular Instruments) pre-hybridisation buffer at 66 °C for 30 min. HCR™ probes for *C9orf72* (GGGGCC)n sense RNA foci were added to hybridisation buffer and sections were incubated overnight at 66 °C in a humidified chamber. Sections were washed 3 times in 50% formamide/50% 0.5x SSC 0.1% tween for 30 min at 66 °C. Sections were then incubated in HCR™ pre-amplification buffer for 30 min at room temperature. Corresponding HCR™ amplifiers were added to the amplification buffer and incubated for 2–2.5 h at room temperature. Sections were washed in 2x SSC and 0.1% Tween for 5 min, twice for 15 min and again for 5 min. Sections were coverslipped with ProLong™ Gold Antifade mounting media with DAPI (P36931, Thermo Fisher Scientific). For Iba1

and GFAP immunohistochemistry, 5 μm-thick sections were deparaffinized in xylene and rehydrated through a series of ethanol solutions, followed by antigen retrieval for 15 min in 1x Citrate Buffer (420902, BioLegend) at 95 °C. Endogenous peroxidase activity was blocked for 10 min in 3% hydrogen peroxide solution at room temperature. Sections were blocked in 10% non-fat dry milk for 30 min. Sections were immunostained with primary antibody against Iba1 or GFAP at a 1 in 500 dilution in blocking solution (GFAP AB5804, Abcam; Iba1 019-19741, FUJIFILM Wako Chemicals USA Corporation) overnight at 4 °C, followed by PBS washes and secondary biotin-conjugated anti-rabbit antibody incubation (BA-1000, Vector Laboratories) for 30 min at RT. Sections were incubated in Vectastain Elite ABC (PK-6100, Vector Laboratories) for 30 min at room temperature, followed by ImmPACT DAB (SK-4105, Vector Laboratories) for 3–5 min. Sections were counterstained with haematoxylin (760-2021, Roche Diagnostics), dehydrated through a series of ethanol and xylene washes, and coverslipped using DPX mounting medium (06522, Sigma Aldrich). For the 2xNLS-mApple immunohistochemistry, brains were embedded in OCT (Tissue-Tek), and 20 μm-thick sections were cut, mounted, and stored at −80 °C until use. To visualise mApple expression, slides were placed at 37 °C for 10 min; washed 3 times with PBS; blocked for 2 h with blocking solution containing normal goat serum (Merck), 5% BSA (Sigma), and 0.2% Triton X-100 (Sigma); and incubated with primary antibodies (RFP ROCK600-401-379, VWR, 1:1000; NeuN ABN91, Sigma, 1:500; GFAP 13-0300, Invitrogen, 1:500) overnight at 4 °C. The next day, slides were washed 3 times with PBS, incubated with secondary antibodies (Gt anti-Rb AlexaFluor546 A-11035, Invitrogen, 1:1000; Gt anti-Ck AlexaFluor488 A-11039, Invitrogen, 1:1000; Gt anti-Rt AlexaFluor633 A-21094, Invitrogen, 1:1000) for 2 h at room temperature, washed again, and coverslipped with Pro-Long Gold antifade mounting media containing DAPI (Invitrogen).

### Microscopy and quantification

Dual RNA-FISH and immunocytochemistry images were acquired using the automated Opera Phenix™ high-throughput confocal imaging platform (Perkin Elmer) and analysed using Columbus 2.8 (PerkinElmer). RNA FISH load was determined by calculating the integrated intensity of nuclear RNA puncta by multiplying spot intensity by total spot load per CasRx positive cell (as determined by nuclear HA positivity). RNA foci in mouse brain sections were imaged with the Leica Mica platform at 63× in widefield mode. One 2 × 2 panelled image was acquired in z-stack per section with equal settings for each animal. Quantification of RNA foci load was done in ImageJ with a custom macro. In brief, after converting each z-stack into a max projection image and post-processing each channel with the Mica THUNDER tool, a mask was created based on the DAPI image and each cell was defined as a separate region of interest (ROI). RNA foci were then identified within the DAPI-positive ROIs and foci load was calculated as average number of foci per ROI and percentage of ROIs containing at least one RNA focus. The average number of DAPI-positive ROIs per panelled image was 218. Iba1 and GFAP-stained slides were scanned with a Zeiss AxioScan Z1 microscope at 20× magnification. Quantification of Iba1 and GFAP percentage area was done in ImageJ with a custom macro. In brief, brain regions of interest were isolated, and a mask was created based on GFAP+ or Iba1+ cells with adjusted threshold to omit the background. GFAP+ or Iba1+ area was calculated as well as total area of cortex or hippocampus. All images were processed using identical settings for background subtraction and thresholding. Area of GFAP+ or Iba1+ cells was presented as a percentage of total brain region area. mApple-stained mouse brain sections were imaged with identical settings using the Leica Mica platform at 20x in widefield mode. A tiled z-stack was acquired across the entire cortex and hippocampus and converted into a max projection image for further analysis. Number of mApple+ cells was calculated as a percentage of DAPI+ cells in each notated brain region using a custom ImageJ macro. In brief, regions of

damage were first cropped from each image in all channels. Then ROIs were chosen for each brain region and number of mApple+ or DAPI+ cells was calculated in each ROI using equal thresholds for all images. The average percentage of DAPI-positive objects was calculated from 3 sections per brain and used for further analysis.

### Statistics and reproducibility

Biologically independent experiment replicates (N number) are indicated in figure legends. Technical replicates are displayed on graphs as well biological replicates. For in vivo experiments, biological replicates were defined as individual mice. No statistical method was used to pre-determine sample sizes. No data were excluded from the analyses. Investigators were blinded to genotype and condition during monthly weight and health assessments and unblinded thereafter for molecular analyses. Statistical analyses throughout the study were performed with GraphPad Prism on biological replicates only, and the tests used are listed in the figure legends. Workflow diagrams made with permissioned use from Biorender.com (publication licenses to A. Cammack, 2024). Volcano plots made with ggVolcanoR[79].

### Reporting summary

Further information on research design is available in Nature Portfolio Reporting Summary linked to this article.

## Data availability

All data in the main text and supplementary materials, are available upon request and are provided as source data files. RNA-seq data is available on GEO, accession GSE255022. Summary RNA-seq data are provided in Supplementary Data File 1. Source data are provided with this paper.

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

## Acknowledgements

We thank Dr. Jamie Evans for technical assistance with FACS, Drs Siddharthan Chandran and Bhuvaneish Selvaraj for providing *C9orf72* patient lines, Dr. Michael Ward for providing the piggyBac NGN2 constructs and Dr. Dieter Edbauer for providing anti-GA monoclonal antibody. Funding: Leonard Wolfson Centre for Experimental Neurology, Leonard Wolfson PhD Programme in Neurodegeneration (L.K., F.K.). Neurogenetic Therapies Programme, funded by Sigrid Rausing Trust (AMI, L.K.). European Research Council, European Union's Horizon 2020 research and innovation programme grant 648716–C9ND (A.M.I.). UK Dementia Research Institute, funded by UK Medical Research Council, Alzheimer's Society, and Alzheimer's Research UK (A.M.I.). EMBO Postdoctoral Fellowship (A.J.C.). Live Like Lou Foundation Postdoctoral Fellowship (A.J.C.). Wolfson-Eisai Neurodegeneration University College London PhD programme, funded by Eisai (R.C.). Alzheimer's UK grant ARUK-DC2021-039 (T.M.R., A.M.). Wellcome Investigator Award 217150/Z/19/Z (J.R.). A.A.R. is funded by the UK Medical Research Council (MR/N026101/1, MR/T044853/1, MR/X004724/1, MR/R025134/1, MR/R015325/1, MR/S009434/1, the Wellcome Trust Institutional Strategic Support Fund/UCL Therapeutic Acceleration Support (TAS) Fund (204841/Z/16/Z), the Sigrid Rausing Trust and the Jameel Education Foundation. A.A.R. is supported by the NIHR Great Ormond Street Hospital Biomedical Research Centre (562868).

## Author contributions

L.K., D.V., A.J.C., and A.M.I. planned and oversaw all aspects of the study. L.K. and D.V. performed and analyzed most of the in vitro experiments. A.J.C. and M.C. performed and analyzed the in vivo mouse experiments. A.N.C. performed and analyzed the iPSC-SN glutamate excitotoxicity and cryptic exon experiments. A.M. analyzed the off-target RNA-seq data. A.F., P.S., B.M., B.V.H., E.K., and R.C. assisted in in vitro experiments. A.McG. and T.M.R. developed and provided the *Tg(ubi:G4C2x45)* zebrafish. F.K. and J.R. performed and analyzed the zebrafish experiments. M.J.R., T.G.M., L.Y., S.V., and P.d.O. assisted in mouse experiments. F.C.M., K.J., L.M.D., and L.P. generated and provided AAVs for this study. Y.Z., L.V.D.B., G.L., A.A.R., A.N.C., J.R., and A.M.I. provided oversight and planning for the studies. L.K., D.V., A.J.C., and A.M.I. wrote the manuscript with input from all authors.

## Competing interests

L.K. and A.M.I. are the co-inventors of a patent filed by UCL Business Ltd on CasRx/Cas13d systems targeting C9orf72, currently at PCT stage, number PCT/EP2022/060296. All other authors declare they have no competing interests.
