## [Transparent Peer Review file · Nature Communications]

Dual-targeting CRISPR-CasRx reduces C9orf72 ALS/FTD sense and antisense repeat RNAs in vitro and in vivo

Corresponding Author: Professor Adrian Isaacs

Editorial Note: Reviewers #2 and #3 collaborated in reviewing this manuscript, and their reports should be regarded as a single, combined assessment.

Version 0:

Reviewer comments:

Reviewer #1

(Remarks to the Author)

In this paper, genetic cause of the ALS FTD disorder which is the expansion of the repeats (G4C2) in the intron 1 of chromosome 9 open reading frame 72 gene transcripts are targeted by Cas13 from *Ruminococcus flavefaciens* named CasRX. It is noteworthy that, both sense and anti-sense transcripts were targeted and distinct dipeptide proteins derived from repeat associated non-AUG translation were also included. It is a strong point in this paper that antisense RNAs and dipeptide protein transcripts were targeted (most of the papers include only sense RNAs).

Based on CRISPR-Cas methodology, this research paper includes the proper guide RNA selection out of gRNA pool by testing with NLuc/Fluc reporter assay. Additionally, it is an important point that two different orthologs of the Cas13 enzyme were tested and the more effective and smaller variant CasRx was decided for further experiments. Effectiveness of the CasRx confirmed in HEK cells, C9orf72 patient-derived iPSC-neuron lines, mouse models and zebra fish. Diversity of models provides comprehensive analysis.

In terms of topic and content, ALS FTD disorder and therapeutic approaches with CRISPR-Cas RNA targeting enzymes are quite hot topic however it is not ground breaking or area opening. There are similar approaches discussed in the literature where as to cure this disease CRISPR based therapeutic approaches are important for the future.

There are specific points regarding this paper that I believe it will improve the research paper in terms of readability and scientific reasoning.

-crRNA maturation was confirmed in the cell lines whereas these informations are redundant for the literature. However, it provides further confirmation and allows for multiplexing approaches for cocktail guide RNA and supplementary Figure 1 with dual guides are actually nice results! Therefore maybe authors might consider to relocate Figure 1S-G&H to main figures and would like to discuss their therapeutic practicality in discussion.

-MAJOR POINT : It is a very big claim that significant off-target effect was not observed!. Most of the CRISPR-Cas enzymes with collateral effect demonstrate off-target effect or toxicity! I would definitely apply Deep RNA-Seq protocol to not only iPSC cells but also all the cells and models being tested in the study!!! This claim and effect is absolutely very important and big claim for the field even though there are supporting literature and evidences! Please do provide more supporting evidences for this claim.

RNA Seq results and lists need to be transparently included as a supplementary document and excel sheets needs to be provided with the analysis otherwise it is difficult to believe the claim!!!

- Some figures have window lines (eg: Figure 2g)

- In order to use correct terminology for CRISPR-Cas: authors might consider to use "collateral activity" instead of "bystander effect".

Figure 1E : Mis-spelling correction: AntiEnse guides

More details could be provided for this study : It will nice to provide guideRNA oligo lists and the vector being cloned and used.

Above critics and comments are aimed to improve the study and keep the high quality of the research paper. I certainly believe that above critics and comment are relatively easy and straightforward to accomplish.

Reviewer #2

(Remarks to the Author)

The study in the manuscript titled "Dual-targeting CRISPR-CasRx reduces C9orf72 ALS/FTD sense and antisense repeat RNAs in vitro and in vivo" by Isaacs research group focuses on evaluating the therapeutic efficacy of CRISPR-CasRx to address the pathology associated with C9orf72 repeat expansion, a common genetic factor in Amyotrophic Lateral Sclerosis (ALS) and Frontotemporal Dementia (FTD). The initial investigations involved in vitro experiments using HEK293 cells to target sense G4C2 repeat-containing transcripts. Thirty-nucleotide guide RNAs (gRNAs) were meticulously designed, selecting six with low homology to other human transcripts. Employing a NanoLuciferase (NLuc) reporter assay, CasRx demonstrated remarkable efficiency, achieving a 99% reduction in NLuc signal with four out of six tested guides. Importantly, the smaller size of CasRx renders it suitable for potential therapeutic delivery through adeno-associated viruses (AAVs). Building on the success of targeting sense repeats, the study further explored the capability of CRISPR-CasRx to address antisense repeat-containing transcripts. An antisense NLuc reporter was engineered, and four gRNAs were designed for targeting the antisense C4G2 repeats. Despite uncertainties about the antisense transcriptional start site, all tested guides exhibited efficacy, reducing the NLuc signal by over 70%. The most effective guide, number 11, achieved an impressive 89% reduction, even in a region with high GC-content (~80%). To validate the findings and confirm the reduction in C9orf72 transcripts, the study employed repeat-targeted RNA fluorescent in situ hybridization (FISH). This approach directly confirmed the lowering of both sense and antisense C9orf72 repeat RNAs to near-background levels. The dual strategy involving the NLuc reporter assays and FISH collectively underscored the robust capability of CasRx in degrading C9orf72 repeat RNAs, preventing the accumulation of dipeptide repeat proteins (DPRs) in vitro.

The study then progressed to assess the translational potential of CRISPR-CasRx in patient-derived neurons. Using induced pluripotent stem cells (iPSCs) from ALS/FTD patients with C9orf72 repeat expansion, an inducible system was established, allowing the rapid differentiation of iPSCs into cortical neurons. Transduction with CRISPR-CasRx and specific gRNAs resulted in significant reductions (30-80%) in sense repeat transcripts and (50-95%) in antisense repeat-containing transcripts across multiple patient iPSC-neuron lines. Importantly, this reduction correlated with a substantial (~60-70%) decrease in polyGA and polyGP DPRs. To address safety concerns, the study investigated potential toxicity and off-target effects of C9orf72 repeat-targeting CRISPR-CasRx in patient iPSC-neurons. Five days post-transduction, there was no overt toxicity observed, as evidenced by cell viability assays. Bulk RNA-seq analysis revealed no significant off-target transcriptional changes, supporting the safety profile of CasRx in this context.

Moving beyond cell cultures, the study tested CRISPR-CasRx in vivo using zebrafish larvae and mouse models. In zebrafish larvae expressing pure G4C2 repeats, CasRx successfully rescued larval hyperactivity, a behavioral phenotype associated with C9orf72 repeats. In mouse models, CasRx effectively targeted sense repeats in (G4C2)₁₄₉ repeat mice, achieving a >50% reduction in sense repeat-containing transcripts. Furthermore, a dual-targeting CRISPR-CasRx AAV in C9orf72 BAC mice simultaneously reduced both sense and antisense repeat transcripts (~20%) and showed variable but overall reduced DPR levels.

While the study extensively emphasizes the therapeutic promise of CRISPR-CasRx in tackling the intricate pathology linked to C9orf72 repeat expansion and demonstrates success in diverse in vitro and in vivo models, there is a need for a thorough reevaluation of this manuscript. Additional experiments are essential to validate the proposed approach effectively. Following key considerations should be addressed to enhance the effectiveness of this approach.

Major concerns:

1. Amyotrophic lateral sclerosis (ALS) is typically late-onset disease, making neonatal treatment strategies are less promising. The C9-500 transgenic mice, specifically the FVB/NJ-Tg(C9orf72)₅₀₀ strain (RRID: IMSR JAX 029099), exhibit a strong phenotype characterized by motor dysfunction and a shortened lifespan (PMID: 27112499). To assess the effectiveness of the suggested therapeutic approach, it is crucial to incorporate behavioral and survival studies.

2. Different cell types exhibit unique expression levels and patterns of C9orf72 isoforms, suggesting that targeting specific cells may result in diverse impacts. The manuscript lacks a comprehensive approach covering both the central nervous system (CNS) and non-CNS tissues, potentially limiting its applicability. To address this, it is recommended to conduct post-treatment assessments of vital organ tissues, such as the liver and lungs.

3. Imaging of CNS cell types? Transduction and tropism into cortical neurons. FISH or DPR image quantification. Microgliosis?

4. The authors major message is the value of including antisense targeting gRNAs for treating C9orf72 ALS/FTD. However, there crucial experiments missing are demonstrating the separate and combined effects of sense and antisense targeting gRNAs (guides 10 and 17). What are the therapeutic efficacies of treating in vivo models with the single or dual guides, and is there a synergistic effect?

Other concerns:

1. Does the potential gRNA selectively downregulate the pathogenic allele? If so, how does the NLuc/FLuc reporter system represent the heterozygous condition? Include a repeat expansion reporter that represents healthy repeat lengths (2-24).

2. Additional details about lentiviral delivery, such as the Multiplicity of Infection (MOI), should be included for a more comprehensive understanding.

3. Are neurons derived from iPSCs within a 5-day differentiation period mature enough to exhibit RNA foci and DPR?

Consider extending the differentiation time and conducting RNA FISH for a more insightful assessment of system efficacy.

4. For Figures 2G a S3E-G, it would be informative to show a diagram of which sequences within the C9orf72 gene are used as the reference for quantifying the RNA-seq transcripts.
5. What are the allelic sizes of the 'G4C2'-repeats in iPSC lines?
6. How can the results in Figure 3B and 3C be correlated and explained in the context of RNA-Protein relation? assayed hippocampal tissue... don't know how well AAV transduced... would this result suggests polyGP comes from antisense transcript, which they did not target in these mice.
7. Figure 4G does not appear very convincing, as most the values in the control group match the values as in the treatment condition. Provide additional clarification or data to support the results.
8. Given that disease onset in C9-500 mice typically occurs around 8 weeks when RNA foci become observable (PMID: 27112499), could you provide the rationale for selecting P0 mice for the treatment? Probably more convenient delivery of AAV? Do these mice exhibit foci in the NT condition and reduced foci in the treated condition?
9. In the mouse treatment experiments, the inclusion of a wild-type control is crucial for comparing post-treatment C9orf72 expression levels and assessing the effectiveness of the treatment. Overall expression of C9orf72 is reduced in C9 patients. Does treatment result in increased expression of Variant 2, and how would this compare to WT levels?
10. Since PiggyBac integration is random, it is advisable to include karyotyping in the study to demonstrate the preservation of intact genomic integrity. Is this typically done in these targeted iPSC lines?

In summary, this manuscript needs to address several concerns before convincingly demonstrating the impact of the CRISPR-Cas13 platform as a viable treatment for C9ORF72 neuropathies.

Reviewer #3

(Remarks to the Author)

Reviewer #4

(Remarks to the Author)

In this manuscript, Kempthorne and colleagues exploited CRISPR-Cas13 systems to target and reduce both the sense and anti-sense repeat RNA species in C9orf72 ALS/FTD in various models. The work is well done, proper controls are included, and the writing is clear. Although the results of this study are of great interest, the manuscript is missing some key data that hampers the conclusions that can be drawn from the experiments performed particularly in terms of the therapeutic utility in the context of ALS/FTD.

1) It is clear from the findings that CasRx can achieve robust reduction in sense and antisense repeat RNAs and DPRs. Can phenotypes associated with ALS/FTD be ameliorated/restored with CRISPR-CasRx? Assays could be performed in the various models to assess the therapeutic potential of the CRISPR-CasRx in more depth.

2) The variability in the efficacy of both sense and anti-sense targeting by CRISPR-CasRx is considerably big. It also depends on the cell line used. What does the difference in the efficacy of the CasRx to target sense and anti-sense RNA mean in the context of different cell lines? What would be its clinical relevance?

3) The article emphasizes the importance of targeting anti-sense repeats and the production of DPRs from the anti-sense. Any proof of the reduction in DPRs generated from anti-sense repeats in patient derived iPSC-neurons will consolidate the findings.

4) Can CRISPR-CasRx effectively targets sense and antisense C9orf72 repeat RNAs in patient-derived iPSC-motoneurons?

5) C9 line 2 had the most reduction in antisense contain repeats transcripts which was considerably different from C9 cell lines 1 and 3. It may be important to show the non-toxic effects in cell line 2.

6) The hyperactivity behavioural phenotype in the zebrafish model expressing 45xG4C2 repeats is very intriguing. Such phenotype was not reported in other published zebrafish models expressing C9orf72 HRE (Swinnen et al., 2018 and Shaw et al., 2018). Assessing motor axonal defects in the zebrafish model expressing 45xG4C2 repeats and potential rescue with CRISPR-CasRx would be clinically more relevant to ALS.

7) Figure 1E, "antiense" should be corrected to "antisense".

Version 1:

Reviewer comments:

Reviewer #1

(Remarks to the Author)

Paper named "Dual-targeting CRISPR-CasRx reduces C9orf72 ALS/FTD sense and antisense repeat RNAs in vitro and in vivo" responded well to the critical points that I stated.

Necessary experiments and documentation are well established after the revision. Necessary typo mistakes and major critics are responded in an organised and well established way.

I don't have any more critics. I think paper is ready for publication scientifically.

Reviewer #2

(Remarks to the Author)

Reviewer #3

(Remarks to the Author)

In this revised manuscript: "Dual-targeting CRISPR-CasRx reduces C9orf72 ALS/FTD sense and antisense repeat RNAs in vitro and in vivo", Kempthorne, Vaizoglu, Cammack, et al. have responded to reviewers' concerns, and many of these have been addressed through additional experiments, clarifications, and modifications to the manuscript.

Topically, treatments for ALS/FTD through gene therapy are of high interest and importance, and this remains an unmet need. This manuscript aims to explore CRISPR-Cas tools and AAV serotype delivery to the CNS to demonstrate principle and efficacy of a potential therapeutic avenue. The dual-targeting CRISPR-CasRx offers some advantages over other CRISPR-Cas systems. Since the Cas13 proteins can mature gRNA arrays as well as degrade their targeted RNA, multiple gRNAs can be included in the same construct. This makes it an attractive candidate to simultaneously target the sense and antisense C9orf72 repeat expansion transcripts. Of course, the motivation to pursue this approach necessitates demonstration that sense and antisense strands meaningfully contribute to disease pathogenesis. Furthermore, CasRx is smaller than other Cas proteins. Altogether, this system is amenable to efficient packaging into AAV vectors, which are known to be size restricted compared to other viruses, and more efficient production of AAVs suggest more efficient means of in vivo transduction of brain and spinal cord cells. Therefore, the other onus is to demonstrate this feature of the dual-targeting CasRx system.

A positive principle of this work is that the manuscript aims to investigate the efficacy of the dual-targeting CasRx system to reverse disease phenotype across multiple in vitro and in vivo models.

The HEK293T data demonstrate a clever in-frame reporter system for sense (polyPR) and antisense (polyGR) expression and a means to identify candidate gRNA sequences that silence them using CasRx.

The iPSC-neurons demonstrate the efficacy of their dual-guide CasRx system to 1) effectively silence endogenous expression of the sense and antisense C9orf72 repeat expansion transcripts, 2) demonstrate that both sense and antisense targeting RNAs can rescue glutamate toxicity individually or together, and 3) attribute the distinct roles of antisense transcripts to cryptic exon inclusion. Overall, these experiments used the dual-targeting CasRx system to provide more insight into the pathogenic mechanisms of the sense and antisense transcripts. As a minor note, since the glutamate toxicity assays were quantifications of imaging data of PI, presumably co-localized with neuronal markers, the authors should present images of the iPSC-SNs, since this is a completely different protocol and cell type from the inducible NGN2 cortical neurons in Figure 2.

In vivo, the authors demonstrate the ability of the dual-guide CasRx system to reverse the hyperactivity phenotype in zebrafish larvae. Since both gRNAs were included in these assays, it is not clear if either or both sense and antisense transcripts were responsible for the phenotype. This is a missed opportunity to dissect their roles in this system and further underscore the importance of targeting both sense and antisense transcripts as a therapy.

In the preceding experimental systems, the dual-guide CasRx system was delivered to the cells and larvae using transfection, lentivirus, nucleofection, or injection. However, gene therapies for neurodegenerative disorders like ALS face important challenges in efficiently and specifically delivering genes to CNS regions and cell types, and this challenge is evident in the mouse data presented by the authors here.

The authors tested the ability of the dual-guide CasRx system to silence sense and antisense transcripts in two mouse models of C9orf72 ALS. One is the 149R mouse, where CasRx PHP.eB AAV and 149R C9orf72 AAV9 were co-injected via ICV into P0 C57BL6/J mice. After three weeks, they observed significantly reduced sense transcript but no reduction in polyGP. The authors also tested their CasRx system in another mouse model: the C9-500 BAC in the FVB/NJ background, also injected via ICV with CasRx PHP.eB AAV at P0. At 8.5 months, they observed reduced sense and antisense transcripts, reduced polyGP but not polyGA, and no reduction of sense RNA foci in the cortex.

These positive and negative results were difficult to interpret and are thus unconvincing without knowing the efficacy of AAV transduction. Therefore, the authors were asked to provide data demonstrating the transduction efficiency and tropism of their AAV delivery approach. The authors have repeated ICV infection of AAV.PHP.eB using a 2xNLS-mApple reporter, and they have concluded that 5-10% of cells across the brain are transduced. In light of the newly added transduction efficiency data, the practical application of this approach as an effective gene therapy candidate remains unknown. This situation

presents a technical challenge that is surmountable, given the widespread use of PHP.eB AAV to deliver genes through intravenous routes, and has been noted to work especially well in the permissive FVB/NJ strain PMID: 31725765. Without effective AAV transduction in the CNS, it remains difficult to demonstrate that the dual-CasRx vector can meaningfully rescue disease pathogenesis in C9orf72 ALS mouse models. Overall, enthusiasm for and impact from this aspect of the manuscript is dampened without more conclusive data in the mouse models. As is, this work as a whole offers strong proof of principle but does not yet offer suitable impact expected from readers of this journal.

Reviewer #4

(Remarks to the Author)

I have no further comments. My concerns have been adequately addressed in the revised version.

We thank the reviewers for their comments which we feel have greatly strengthened the paper. Below we address each comment point-by-point.

Reviewer #1 (Remarks to the Author):

In this paper, genetic cause of the ALS FTD disorder which is the expansion of the repeats (G4C2) in the intron 1 of chromosome 9 open reading frame 72 gene transcripts are targeted by Cas13 from *Ruminococcus flavefaciens* named CasRx. It is noteworthy that, both sense and anti-sense transcripts were targeted and distinct dipeptide proteins derived from repeat associated non-AUG translation were also included. It is a strong point in this paper that antisense RNAs and dipeptide protein transcripts were targeted (most of the papers include only sense RNAs).

Based on CRISPR-Cas methodology, this research paper includes the proper guide RNA selection out of gRNA pool by testing with NLuc/Fluc reporter assay. Additionally, it is an important point that two different orthologs of the Cas13 enzyme were tested and the more effective and smaller variant CasRx was decided for further experiments.

Effectiveness of the CasRx confirmed in HEK cells, C9orf72 patient-derived iPSC-neuron lines, mouse models and zebra fish. Diversity of models provides comprehensive analysis.

In terms of topic and content, ALS FTD disorder and therapeutic approaches with CRISPR-Cas RNA targeting enzymes are quite hot topic however it is not ground breaking or area opening. There are similar approaches discussed in the literature where as to cure this disease CRISPR based therapeutic approaches are important for the future.

There are specific points regarding this paper that I believe it will improve the research paper in terms of readability and scientific reasoning.

- crRNA maturation was confirmed in the cell lines whereas these informations are redundant for the literature. However, it provides further confirmation and allows for multiplexing approaches for cocktail guide RNA and supplementary Figure 1 with dual guides are actually nice results! Therefore maybe authors might consider to relocate Figure 1S-G&H to main figures and would like to discuss their therapeutic practicality in discussion. Move 1SG-H to main figure 1

We thank the reviewer for raising this point. We have added a line to the discussion to highlight this and have swapped original Figure 1C-E and Figure 1SE-H as the reviewer suggests.

- MAJOR POINT : It is a very big claim that significant off-target effect was not observed!. Most of the CRISPR-Cas enzymes with collateral effect demonstrate off-target effect or toxicity! I would definitely apply Deep RNA-Seq protocol to not only iPSC cells but also all the cells and models being tested in the study!!! This claim and effect is absolutely very important and big claim for the field even though there are supporting literature and evidences! Please do provide more supporting evidences for this claim. RNA Seq results and lists need to be transparently included as a supplementary document and excel sheets needs to be provided with the analysis otherwise it is difficult to believe the claim!!!

We agree it was surprising that we did not observe off-targets, and we think there are likely to be ones that would appear with more sensitive measures. We have therefore edited the text in the abstract, results, and discussion to avoid over-stating this data and clarify that this was only seen with a standard read depth (20 million reads per sample) and that more sensitive measures would likely identify some off-targets.

As requested, the raw and processed RNA-seq data from the iPSC-neurons have been uploaded to GEO, accession GSE255022 with private reviewer access via token (ctslekmmlynmv). The GEO file will be made publicly available upon publication of the paper. For convenience, we include with this revision an excel file (Supplementary Data File 1) with fold changes and significance for all genes in all comparisons from these analyses.

- Some figures have window lines (eg: Figure 2g)

We are not clear why this is the case as there are not window lines on our version.

- In order to use correct terminology for CRISPR-Cas: authors might consider to use "collateral activity" instead of "bystander effect".

Thank you for pointing this out. We have now changed bystander effect to collateral activity as suggested.

- Figure 1E : Mis-spelling correction: AntiEense guides

Thank you for spotting this, we have now corrected it.

- More details could be provided for this study : It will nice to provide guideRNA oligo lists and the vector being cloned and used.

We apologise that the gRNA sequences were not provided, this was an oversight. We have now added Table 3 with the gRNA sequences and additional cloning information in the methods.

Reviewer #2 (Remarks to the Author):

The study in the manuscript titled "Dual-targeting CRISPR-CasRx reduces C9orf72 ALS/FTD sense and antisense repeat RNAs in vitro and in vivo" by Isaacs research group focuses on evaluating the therapeutic efficacy of CRISPR-CasRx to address the pathology associated with C9orf72 repeat expansion, a common genetic factor in Amyotrophic Lateral Sclerosis (ALS) and Frontotemporal Dementia (FTD). The initial investigations involved in vitro experiments using HEK293 cells to target sense G4C2 repeat-containing transcripts. Thirty-nucleotide guide RNAs (gRNAs) were meticulously designed, selecting six with low homology to other human transcripts. Employing a NanoLuciferase (NLuc) reporter assay, CasRx demonstrated remarkable efficiency, achieving a 99% reduction in NLuc signal with four out of six tested guides. Importantly, the smaller size of CasRx renders it suitable for potential therapeutic delivery through adeno-associated viruses (AAVs).

Building on the success of targeting sense repeats, the study further explored the capability of CRISPR-CasRx to address antisense repeat-containing transcripts. An antisense NLuc reporter was engineered, and four gRNAs were designed for targeting the antisense C4G2 repeats. Despite uncertainties about the antisense transcriptional start site, all tested guides exhibited efficacy, reducing the NLuc signal by over 70%. The most effective guide, number 11, achieved an impressive 89% reduction, even in a region with high GC-content (~80%). To validate the findings and confirm the reduction in C9orf72 transcripts, the study employed repeat-targeted RNA fluorescent in situ hybridization (FISH). This approach directly confirmed the lowering of both sense and antisense C9orf72 repeat RNAs to near-background levels. The dual strategy involving the NLuc reporter assays and FISH collectively underscored the robust capability of CasRx in degrading C9orf72 repeat RNAs, preventing the accumulation of dipeptide repeat proteins (DPRs) in vitro.

The study then progressed to assess the translational potential of CRISPR-CasRx in patient-derived neurons. Using induced pluripotent stem cells (iPSCs) from ALS/FTD patients with C9orf72 repeat expansion, an inducible system was established, allowing the rapid differentiation of iPSCs into cortical neurons. Transduction with CRISPR-CasRx and specific gRNAs resulted in significant reductions (30-80%) in sense repeat transcripts and (50-95%) in antisense repeat-containing transcripts across multiple patient iPSC-neuron lines. Importantly, this reduction correlated with a substantial (~60-70%) decrease in polyGA and polyGP DPRs. To address safety concerns, the study investigated potential toxicity and off-target effects of C9orf72 repeat-targeting CRISPR-CasRx in patient iPSC-neurons. Five days post-transduction, there was no overt toxicity observed, as evidenced by cell viability assays. Bulk RNA-seq analysis revealed no significant off-target transcriptional changes, supporting the safety profile of CasRx in this context.

Moving beyond cell cultures, the study tested CRISPR-CasRx in vivo using zebrafish larvae and mouse models. In zebrafish larvae expressing pure G4C2 repeats, CasRx successfully rescued larval hyperactivity, a behavioral phenotype associated with C9orf72 repeats. In mouse models, CasRx effectively targeted sense repeats in (G4C2)₁₄₉ repeat mice, achieving a >50% reduction in sense repeat-containing transcripts. Furthermore, a dual-targeting CRISPR-CasRx AAV in C9orf72 BAC mice simultaneously reduced both sense and antisense repeat transcripts (~20%) and showed variable but overall reduced DPR levels.

While the study extensively emphasizes the therapeutic promise of CRISPR-CasRx in tackling the intricate pathology linked to C9orf72 repeat expansion and demonstrates success in diverse in vitro and in vivo models, there is a need for a thorough reevaluation of this manuscript. Additional experiments are essential to validate the proposed approach effectively.

Following key considerations should be addressed to enhance the effectiveness of this approach.

Major concerns:

- Amyotrophic lateral sclerosis (ALS) is typically late-onset disease, making neonatal treatment strategies are less promising. The C9-500 transgenic mice, specifically the FVB/NJ-Tg(C9orf72)500 strain (RRID: IMSR JAX 029099), exhibit a strong phenotype characterized by motor dysfunction and a shortened lifespan (PMID: 27112499). To assess the effectiveness of the suggested therapeutic approach, it is crucial to incorporate behavioral and survival studies.

Our colony of C9-500 mice do not exhibit motor dysfunction or a shortened lifespan. Most other labs have also not observed this phenotype, and this was published in *Neuron: Absence of Survival and Motor Deficits in 500 Repeat C9ORF72 BAC Mice*. Mordes DA et al *Neuron*. 2020 Nov 25;108(4):775-783.e4. Therefore it is not possible to rescue a phenotype which is not present in our colony. The behaviour phenotype of these mice is thus rather controversial and may be due to differences in genetic background of the different colonies. However, the C9orf72 field are in consensus that the C9-500 mice are an excellent model for investigating and rescuing C9 pathology and so we have performed a pathology study, which is the best that can be achieved with the current mouse models available.

- Different cell types exhibit unique expression levels and patterns of C9orf72 isoforms, suggesting that targeting specific cells may result in diverse impacts. The manuscript lacks a comprehensive approach covering both the central nervous system (CNS) and non-CNS tissues, potentially limiting its applicability. To address this, it is recommended to conduct post-treatment assessments of vital organ tissues, such as the liver and lungs.

We performed intracerebroventricular (ICV) injections, which purposefully limit the delivery of the virus to peripheral tissues. In addition, in these proof-of-concept experiments we used PHP.eB AAV, which shows almost no transduction of the liver after ICV injection (PMID: 33294493; see figure 5A-B). We apologise that we did not make this clearer and have now added text to make this clearer.

- Imaging of CNS cell types? Transduction and tropism into cortical neurons. FISH or DPR image quantification. Microgliosis?

We have tried very hard to visualise CasRx to enable these studies, including generating a new antibody against CasRx and attempting single molecule FISH experiments, but unfortunately neither approach was successful. Therefore, to address this question we have analysed the neuronal tropism and transduction of an identical PHP.eB AAV expressing 2xNLS-mApple under the exact same promoter and regulatory sequences with the same titer and delivery method. We show that transduction efficiency in this paradigm is only ~5-10%, depending on the brain region, and that transduction is largely confined to neurons and astrocytes (see new Figure S8).

We have now performed RNA FISH to visualize sense RNA foci in the BAC mice, however found no change compared to non-targeting control (see new Figure S7). This is unsurprising given the low level of transduction and the high degree of variability in foci load between mice.

We have now performed both IBA1 staining for microgliosis and GFAP staining for astrocytosis and performed image analysis in various brain regions, including the hippocampus, which we show to most highly express PHP.eB AAVs after ICV injection (as also reported by others e.g. PMID: 33294493). We show no change in astrocytosis or microgliosis in cortex or hippocampus upon delivery of CasRx AAVs (see new Figure S6).

- The authors major message is the value of including antisense targeting gRNAs for treating C9orf72 ALS/FTD. However, there crucial experiments missing are demonstrating the separate and combined effects of sense and antisense targeting gRNAs (guides 10 and 17). What are the

therapeutic efficacies of treating in vivo models with the single or dual guides, and is there a synergistic effect?

We agree with the reviewer that it would be very insightful to use our approach to dissect the contribution of sense and antisense repeats to disease pathogenesis. Unfortunately, current in vivo models do not allow us to do this. As discussed above the C9-500 mice do not have a phenotype and therefore we can't show whether single or dual targeting have different or synergistic effects. However, this question is addressable in iPSC-neurons, which display a robust and reproducible glutamate excitotoxicity phenotype (PMID: 32673563, PMID: 24139042) as well as expression of cryptic exons indicative of TDP-43 dysfunction (PMID: 24139042). We have now tested our sense and antisense guides individually and together in this paradigm and found that both are able to independently rescue survival after glutamate stress, while only antisense is able to rescue cryptic exon expression (see new Figures 3 and S4). This result mirrors recently published data showing that only antisense reduction is able to prevent cryptic exon inclusion (PMID: 24139042). We thank the reviewer for suggesting these experiments, which have provided valuable new insights into C9 mechanism and therapeutic targeting.

Other concerns:

- Does the potential gRNA selectively downregulate the pathogenic allele? If so, how does the NLuc/FLuc reporter system represent the heterozygous condition? Include a repeat expansion reporter that represents healthy repeat lengths (2-24).

Our NLuc reporter only expresses NLuc in the presence of an expanded allele as it relies on RAN translation of the expanded repeat. It is therefore not possible to generate a comparable NLuc reporter with a short repeat. However, in order to show specificity for the expanded repeat, we show in patient iPSC-neurons in Figures 2D and S4F,L that our gRNAs do not affect the transcript variant that does not contain the repeat.

- Additional details about lentiviral delivery, such as the Multiplicity of Infection (MOI), should be included for a more comprehensive understanding.

We use lentiviral supernatant directly from our HEK cells, and we have now added this extra detail to the methods. Because we are FACS sorting to select transduced cells, we did not calculate absolute titers for our viruses, but of note the % of cells expressing the GFP reporter was low for all viruses used so MOI is likely <1.

- Are neurons derived from iPSCs within a 5-day differentiation period mature enough to exhibit RNA foci and DPR? Consider extending the differentiation time and conducting RNA FISH for a more insightful assessment of system efficacy.

We thank the reviewer for this suggestion and attempted RNA foci quantification using RNA FISH in our C9 BAC mice. We were initially surprised that we did not observe a lowering of sense RNA foci load in our cohort (see new Figure S7). However, our new results with the matched mApple AAV demonstrated a very low transduction efficiency of only ~5-10% (see new Figure S8) which, along with the high variability of foci load between animals, likely explains the lack of foci reduction.

- For Figures 2G a S3E-G, it would be informative to show a diagram of which sequences within the C9orf72 gene are used as the reference for quantifying the RNA-seq transcripts.

We have added more detail to the methods section about how the C9orf72 repeat and non-repeat containing transcripts were quantified in our RNA-seq analysis. In short, we separated transcripts by presence of either exon 1a (i.e. variants 1 and 3 which contain the repeat) or exon 1b (i.e. variant 2) based on Ensembl-annotated transcript variants, which we have now listed in the methods.

- What are the allelic sizes of the 'G4C2'-repeats in iPSC lines?

We apologise that this was not included, we have now added Table 1 to the methods to include this information and other relevant information for the iPSC lines used in the study.

- How can the results in Figure 3B and 3C be correlated and explained in the context of RNA-Protein relation? assayed hippocampal tissue... don't know how well AAV transduced... would this result suggests polyGP comes from antisense transcript, which they did not target in these mice.

We assessed hippocampus in these animals because previous reports demonstrated that this region most highly expresses AAVs after ICV injection (PMID: 33294493), a result we have now corroborated with our own mApple AAVs (see new Figure S8). We only treated for three weeks, which was sufficient time to

reduce the RNA but may not have been sufficient time to also reduce polyGP due to its longer half-life. Another potential explanation, as the reviewer suggests, is that a proportion of polyGP is derived from the antisense strand. We thank the reviewer for raising this and have now added these potential explanations to the discussion.

- Figure 4G does not appear very convincing, as most the values in the control group match the values as in the treatment condition. Provide additional clarification or data to support the results.

We agree that the effects on DPRs are modest, which is most likely due to us analysing bulk tissue while a minority (~5-10%) of cells within this bulk lysate were transduced. We observe a much greater variance in DPRs than we do in repeat RNA transcripts between animals, perhaps due to variability in extraction of these highly insoluble proteins, thus limiting our ability to see small differences in levels. We have provided this explanation in the discussion.

- Given that disease onset in C9-500 mice typically occurs around 8 weeks when RNA foci become observable (PMID: 27112499), could you provide the rationale for selecting P0 mice for the treatment? Probably more convenient delivery of AAV? Do these mice exhibit foci in the NT condition and reduced foci in the treated condition?

The reviewer is correct that we chose P0 as it was the most convenient method to deliver CasRx to the brain for these proof-of-concept studies. We have now explained this in the discussion. We have also now analyzed RNA foci levels in our CasRx treated BAC mice and we did not observe a lowering of RNA foci load in our cohort, likely because of our low transduction efficiency of only ~5-10% (see new Figure S8) and the high variability of foci load between animals (see new Figure S7).

- In the mouse treatment experiments, the inclusion of a wild-type control is crucial for comparing post-treatment C9orf72 expression levels and assessing the effectiveness of the treatment. Overall expression of C9orf72 is reduced in C9 patients. Does treatment result in increased expression of Variant 2, and how would this compare to WT levels?

We demonstrate in Figure 2, new Figure 3, and Figure 5 that human variant 2 is unchanged upon CasRx delivery *in vitro* and *in vivo*. It is not possible to use a wild-type mouse for measuring the effect on normal C9orf72 levels as we are specifically targeting the human C9orf72 transgene and so the corresponding sequence is not present in wild-type mice. Nonetheless, we have now performed a western blot for total C9orf72 levels in the BAC mice and showed that there is no change upon CasRx delivery (see new Figure S6). However, we note the limitation of this approach, which is that the mouse and human C9orf72 proteins are of identical size and no antibodies currently exist which can selectively identify one or the other. Thus, this western blot quantifies a band which contains both mouse (which is not targeted by CasRx) and human C9orf72.

- Since PiggyBac integration is random, it is advisable to include karyotyping in the study to demonstrate the preservation of intact genomic integrity. Is this typically done in these targeted iPSC lines?

As PiggyBac integration is random we keep the cells as a mixed population after FACS to avoid integration artefacts that could occur in a clonal line. We apologise for not making this clear and have now added this extra detail to the methods.

In summary, this manuscript needs to address several concerns before convincingly demonstrating the impact of the CRISPR-Cas13 platform as a viable treatment for C9ORF72 neuropathies.

Reviewer #4 (Remarks to the Author):

In this manuscript, Kempthorne and colleagues exploited CRISPR-Cas13 systems to target and reduce both the sense and anti-sense repeat RNA species in C9orf72 ALS/FTD in various models. The work is well done, proper controls are included, and the writing is clear. Although the results of this study are of great interest, the manuscript is missing some key data that hampers the conclusions that can be drawn from the experiments performed particularly in terms of the therapeutic utility in the context of ALS/FTD.

- It is clear from the findings that CasRx can achieve robust reduction in sense and antisense repeat RNAs and DPRs. Can phenotypes associated with ALS/FTD be ameliorated/restored with CRISPR-CasRx? Assays could be performed in the various models to assess the therapeutic potential of the CRISPR-CasRx in more depth.

In order to address this point, we have now assessed the effect of CasRx on glutamate excitotoxicity and cryptic exon phenotypes in patient iPSC-derived spinal neurons and demonstrated robust rescues of both phenotypes (see new Figures 3 and S4). These results, along with our previous results showing effectiveness of CasRx at preventing hyperactivity in C9 zebrafish (now Figure S5), demonstrate the therapeutic potential of CasRx-mediated lowering for C9 ALS/FTD.

- The variability in the efficacy of both sense and anti-sense targeting by CRISPR-CasRx is considerably big. It also depends on the cell line used. What does the difference in the efficacy of the CasRx to target sense and anti-sense RNA mean in the context of different cell lines? What would be its clinical relevance?

We agree that we observe variability in the effect size between different cell lines in Figure 2. We think that one reason for this is different levels of lentiviral CasRx expression. Indeed, we show that the extent of antisense repeat reduction is correlated to the level of CasRx within the cell (Figures 2E and S3A). Importantly, in our new Figure 3, in which we used nucleofection rather than lentiviral transduction and used a greater number of patient iPSC lines, we showed a substantially enhanced knockdown which was consistent across the 6 lines tested. Thus, we think the variability observed in Figure 2 is likely due to technical differences in viral expression.

- The article emphasizes the importance of targeting anti-sense repeats and the production of DPRs from the anti-sense. Any proof of the reduction in DPRs generated from anti-sense repeats in patient derived iPSC-neurons will consolidate the findings.

Unfortunately, we are unable to detect the antisense DPRs in patient iPSC-neurons so we are not able to perform this experiment. We have put considerable effort into developing MSD immunoassays for the DPRs but do not yet have the sensitivity required to detect the antisense DPRs in iPSC-neurons.

- Can CRISPR-CasRx effectively targets sense and antisense C9orf72 repeat RNAs in patient-derived iPSC-motoneurons?

We have now performed rescue experiments on glutamate excitotoxicity and cryptic exon phenotypes in C9orf72 patient iPSC-spinal motor neurons and demonstrated robust rescues (see new Figures 3 and S4).

- C9 line 2 had the most reduction in antisense contain repeats transcripts which was considerably different from C9 cell lines 1 and 3. It may be important to show the non-toxic effects in cell line 2.

We have now performed a viability assay in C9 line 2 as suggested and show no significant decrease in cell viability (see new Figure S3D).

- Figure 1E, “antiense” should be corrected to “antisense”.

Thanks for spotting this we have now corrected it.

- The hyperactivity behavioural phenotype in the zebrafish model expressing 45xG4C2 repeats is very intriguing. Such phenotype was not reported in other published zebrafish models expressing C9orf72 HRE (Swinnen et al., 2018 and Shaw et al., 2018). Assessing motor axonal defects in the zebrafish model expressing 45xG4C2 repeats and potential rescue with CRISPR-CasRx would be clinically more relevant to ALS.

We thank the reviewer for raising this interesting point. The Swinnen et al. 2018 paper used injected RNA, not a transgenic fish, so they would not have been able to assess locomotor changes at 5 days post fertilisation as we did, as the RNA would be degraded by then. In any case, they did not look at behaviour. Shaw et al. 2018 used a transgenic approach, but they only measured changes in larval zebrafish activity at light-dark transitions in a 10min-light on, 10min-light off assay. Since this assay reliably induces in wild type larvae a characteristic hyperactivity state called the visual motor response (VMR) or dark photokinesis (Emran et al., 2008), this approach likely would have masked the ability to detect any further hyperactivity in the transgenic line.

We note that hyperactivity is actually a well described phenotype in C9orf72 repeat expansion models, having been previously observed in mouse models over-expressing 66 or 149 G4C2 repeats (PMID:

25977373; PMID: 30767771) and our own *Drosophila* model expressing 36 repeats (PMID: 37733679). The hyperactivity we observe in the zebrafish model is therefore a consistent C9 repeat-related phenotype and one we think adequate to show the point we were trying to make – that CasRx-mediated reduction of the repeats is sufficient to improve a C9-related phenotype. We have now added this point and relevant citations to the discussion.

To see if we can use this model to test early motor axonal defects, as suggested by the reviewer, we first checked if the G4C2 animals have such a defect, following the Swinnen and Shaw assays. We staged (i.e. matched developmental times) the larvae in the morning at 5 hours (shield stage) and again at 24 hours compared SV2-labeled motor axons between G4C2-dsRed fish with WT (dsRed negative) siblings at 30 hpf. Unlike the Swinnen model, we observed no difference in stage and clutch-matched larvae for motor axon growth. While the reasons for these differences are unclear, we hypothesize that axon defects observed in the Swinnen paper could be due to a generalized developmental delay or not matching siblings carefully, as we did observe variation between clutches. We include these data here for reviewer reference but did not include in the main manuscript since we are unable to assess effect of CasRx treatment and therefore do not feel it is pertinent to our study.